# Dynamically expressed genes provide candidate viability biomarkers in a model coccidian

**Matthew S. Tucker¤, Celia N. O'Brien, Mark C. Jenkins, Benjamin M. Rosenthal**[ID]*

United States Department of Agriculture, Agricultural Research Service, Beltsville Agricultural Research Center, Beltsville, MD, United States of America

¤ Current address: Department of Biological Science, Florida Gulf Coast University, Ft. Myers, FL, United States of America
* Benjamin.rosenthal@usda.gov

**Data Availability Statement:** All relevant data are within the manuscript and its Supporting Information files.

**Funding:** This work was supported by USDA Projects "Detection and Control of Foodborne

## Abstract

*Eimeria* parasites cause enteric disease in livestock and the closely related **Cyclospora cayetanensis** causes human disease. Oocysts of these coccidian parasites undergo maturation (sporulation) before becoming infectious. Here, we assessed transcription in maturing oocysts of *Eimeria acervulina*, a widespread chicken parasite, predicted gene functions, and determined which of these genes also occur in *C. cayetanensis*. RNA-Sequencing yielded ~2 billion paired-end reads, 92% of which mapped to the *E. acervulina* genome. The ~6,900 annotated genes underwent temporally-coordinated patterns of gene expression. Fifty-three genes each contributed >1,000 transcripts per million (TPM) throughout the study interval, including cation-transporting ATPases, an oocyst wall protein, a palmitoyltransferase, membrane proteins, and hypothetical proteins. These genes were enriched for 285 gene ontology (GO) terms and 13 genes were ascribed to 17 KEGG pathways, defining housekeeping processes and functions important throughout sporulation. Expression differed in mature and immature oocysts for 40% (2,928) of all genes; of these, nearly two-thirds (1,843) increased their expression over time. Eight genes expressed most in immature oocysts, encoding proteins promoting oocyst maturation and development, were assigned to 37 GO terms and 5 KEGG pathways. Fifty-six genes underwent significant upregulation in mature oocysts, each contributing at least 1,000 TPM. Of these, 40 were annotated by 215 GO assignments and 9 were associated with 18 KEGG pathways, encoding products involved in respiration, carbon fixation, energy utilization, invasion, motility, and stress and detoxification responses. Sporulation orchestrates coordinated changes in the expression of many genes, most especially those governing metabolic activity. Establishing the long-term fate of these transcripts in sporulated oocysts and in senescent and deceased oocysts will further elucidate the biology of coccidian development, and may provide tools to assay infectiousness of parasite cohorts. Moreover, because many of these genes have homologues in *C. cayetanensis*, they may prove useful as biomarkers for risk.

Parasites for Food Safety" 8042-32000-113-00D and "Development of Control and Intervention Strategies for Avian Coccidiosis" 8042-32000-111-00-D.

**Competing interests:** The authors have declared that no competing interests exist.

## Introduction

Coccidian parasites cause significant disease in animals and people. These parasites include species of *Toxoplasma*, *Sarcocystis*, *Neospora*, *Cyclospora*, and *Eimeria*. Coccidiosis, the enteric disease resulting from infection with species of *Eimeria* spp., cost the poultry industry an estimated $14 billion (U.S.) globally each year [1]. Seven species of *Eimeria* infect chickens and each parasitizes specific regions of the intestinal tract. Multiple species of *Eimeria* may coinfect flocks, complicating control efforts. Pathogenicity ranges from mild to severe disease and can include bloody diarrhea and severe inflammatory hemorrhage. Intestinal epithelial damage from coccidiosis can predispose birds to necrotic enteritis caused by *Clostridium perfringens* [2], which is also a common cause of foodborne illness worldwide.

Poultry farmers expend significant resources to limit losses to coccidiosis. Just as *Eimeria* spp. present a significant problem to the poultry industry, the related human parasite *Cyclospora cayetanensis* is emerging as an important food safety threat. Efforts by produce growers to curtail the health impact of *C. cayetanensis* have been hampered by difficulty in studying this enigmatic parasite. Thus, progress in studying the development of closely related coccidian oocysts could yield major benefits to animal production and public health.

Like other species of enteric coccidia, *C. cayetanensis* (first described over 25 years ago [3, 4]) is transmitted by the fecal-oral route. The parasite causes diarrhea, low-grade fever, abdominal cramping, bloating, weight loss, fatigue, and other symptoms [5, 6]. *C. cayetanensis*-contaminated fresh produce such as berries, leafy greens, raspberries, blackberries, mesclun, bagged mixed greens, snow and snap peas, coleslaw, cilantro, and basil have caused numerous foodborne outbreaks worldwide [7]. Transmission, in more than 60 countries, predominates in tropical and subtropical areas, where prevalence can exceed 10% in people [8]. The infection is perennially endemic in impoverished areas where water and food sanitation are poor, and children are at especially high risk [3, 6, 7]. Since the mid-1990s, *C. cayetanensis* has been increasingly recognized as the causative agent of multistate seasonal outbreaks of diarrheal illness in the United States and Canada [6, 7]. In fact, foodborne outbreaks of human cyclosporiasis have recently grown in their extent and cost; the Centers for Disease control and Prevention (CDC) reported ~2,500 cases and 144 hospitalizations in the United States (U.S.) in 2019 [9].

All *Eimeria* and *Cyclospora* species share a common life cycle: environmentally resilient immature oocysts are passed from infected hosts and undergo extrinsic development before becoming infectious. The oocysts of *Eimeria spp*. infecting chickens require oxygen and optimally sporulate at 29°C. Sporulation time varies by species, but sporulation is normally completed in 24–48 hours [10]. In *E. acervulina*, sporulation is typically complete by 22 hours [10]. Oocysts of *C. cayetanensis* may take up to a week to sporulate [3]. Sporulated (mature) avian *Eimeria* oocysts, once ingested, mechanically ruptured, and stimulated, each release four sporocysts. Each sporocyst contains two sporozoites, which are released upon sporocyst excystation. Exposure to bile salts and digestive enzymes hastens this process. Motile sporozoites invade intestinal enterocytes and then undergo asexual development. Sexual development produces oocysts, which are passed into the intestinal lumen and excreted (unsporulated) with feces.

Although the U.S. Food and Drug Administration (FDA) has recently validated a method for diagnosing the occurrence of *C. cayetanensis* in fresh produce [11], the produce industry and their regulators lack the means to determine whether a given cohort of oocysts is infectious. With no animal model and no cell culture means to propagate the human parasite, *in vitro* means to establish viability would provide a critical tool for risk assessment and regulatory action. Furthermore, future interventions might benefit from greater understanding of how such oocysts develop and become infectious. Poultry *Eimeria* serve as a safe and abundant

source of oocysts, offering unparalleled means to elucidate parasite development and seek new, needed methods for risk assessment and control.

Here, we took advantage of a promising model organism (*E. acervulina*), making use of our ability to produce it in abundance. We aimed to understand the fundamental processes governing oocysts maturation and applicability to the human pathogen *C. cayetanensis*. We measured changing gene expression over the first 24 hours of extrinsic development using RNA-Sequencing (RNA-Seq), verifying highly expressed genes during oocyst sporulation by Reverse Transcription quantitative PCR (RT-qPCR). We identified strong candidate biomarkers during sporulation and up-regulated genes demarcating immature and mature *E. acervulina* oocysts. We further determined that many of these genes are also present in other coccidians, including *C. cayetanensis*. We thereby identified genes meriting validation as biomarkers for mature, infectious oocysts in a range of veterinary and human parasites.

## Materials and methods

### Parasites

Three male broiler chickens (Hubbard/Ross HR708, Longneckers Hatchery, Elizabethtown, PA, USA) were infected by oral gavage with 150,000 oocysts of *E. acervulina* (APU-1). Birds were fed a standard poultry starter ration (crumbles, 24% protein) and given water *ad libitum*. On day six post-inoculation, feces from each bird (A, B, C, representing biological replicates) were harvested. Unsporulated *E. acervulina* oocysts were isolated from chicken feces by flotation on saturated salt solution and washed 3–4 times with water. Briefly, feces were collected from individual chickens with a sterile spatula. Feces were resuspended in water and homogenized in a sterile blender. After straining to remove larger debris, oocysts were washed in water and centrifuged at 300 X g for 10 minutes. A saturated NaCl solution (360 g/L) was added to resuspend pellets and float oocysts. After centrifugation, oocysts were collected from the top of the solution and washed repeatedly with water. Oocysts were enumerated by microscopy with a hemocytometer (yielded $10^7$–$10^8$ total oocysts per replicate). The final preparation of oocysts was resuspended in 2% v/v potassium dichromate in a sterile 1-L flask and incubated in a 29˚C shaking water bath. During incubation, the oocyst suspension was aerated with an aquarium pump to promote sporulation. At the beginning of the time course (T0) and other time points (every four hours up to 24 hours [T4, T8, T12, T16, T20, T24]), 50 ml was taken from each replicate flask and a sample was counted with a hemocytometer ($10^6$–$10^7$ oocysts per replicate and time point). In this study, we define parasite cohorts enriched for unsporulated oocysts as immature and cohorts enriched for sporulated oocysts as mature. At each time point, oocysts were centrifuged at 300 X g for 5 min at 4˚C and washed with deionized water to remove excess potassium dichromate. T0 and T24 oocysts were examined at 400X with a Zeiss AxioScope microscope (Zeiss, Germany). Images were captured using Axio-Vision imaging software. Initially, no oocysts were sporulated (S1 Fig). By the 24th hour, 83% +/- 4% of oocysts appeared to have completed sporulation as indicated by the presence, in each, of four distinct sporocysts (S2 Fig). Oocysts were then incubated with sodium hypochlorite (6%) for 15 min with agitation at room temperature to clean them of exogenous microbial contamination. After bleach treatment, samples were diluted with water and the bleach was removed by repeated (4–5 times) centrifugation with water. After washing, Trizol (Thermo Fisher, Waltham, MA, USA) was added to the oocyst pellets and they were frozen at -80˚C.

### Ethics statement

Animal experiments were performed following the protocol (19–04) approved by the Beltsville Area Animal Use and Care Committee, United States Department of Agriculture. Chickens

employed in this study exhibited no outward signs of severe disease over the course of the study. After the study's conclusion, these chickens were euthanized and all efforts were made to minimize animal suffering.

## Total RNA preparation

Total RNA was extracted from frozen oocyst Trizol pellets for each biological replicate by one of two methods. In the first method, frozen oocysts were thawed on ice and 1 ml was transferred to a glass homogenizing tube. Fresh Trizol was added to the oocysts and the suspension ground with a Teflon pestle for five cycles of 25 grinds each. The homogenate was transferred to microcentrifuge tubes containing sterile, Ribonuclease (RNase)-free 0.5 mm glass beads. The tubes were vortexed for 2 min, and centrifuged at 300 X g for 10 minutes at 4˚C. The supernatant was transferred to new tubes, mixed with chloroform, and centrifuged at 12,000 X g for 15 minutes at 4˚C. The organic phase was transferred to new tubes and mixed with an equal volume of 70% ethanol. Total RNA was then isolated using a RNeasy kit (Qiagen, Germantown, MD, USA) following manufacturer's instructions.

The second method of isolation included these modifications: Organic layers from chloroform extraction were mixed with isopropanol and incubated for 10 min at room temperature. Tubes were centrifuged at 12,000 g for 10 min at 4˚C. Pellets were resuspended in 1 ml of 75% ethanol, vortexed, and centrifuged at 7,500 X g for 5 min at 4˚C. Isolated RNA was resuspended in RNase-free water and incubated at 65˚C for 15 min. Total RNA was quantified using a Qubit 3.0 fluorometer (Thermo Fisher, Waltham, MA, USA) and quality was assessed using a Bioanalyzer 2100 (Agilent Technologies, Santa Clara, CA, USA). RNA samples were frozen at -80˚C. Prior to RNA-Seq, DNA was removed from total RNA with a Turbo DNA-free kit (Thermo Fisher, Waltham, MA, USA). RNA quantity and quality were re-assessed by Qubit and Bioanalyzer. Total RNA of suitable quality for RNA-Seq had an RNA Integrity Number (RIN) ≥7 when assessed by a Bioanalyzer 2100.

## cDNA library construction and RNA-Seq

Whole coding transcriptome libraries were produced from 2–3 biological replicates of *E. acervulina* oocysts collected at each time point. Approximately 200 ng of total RNA was used as input with the TruSeq Stranded mRNA kit (Illumina, San Diego, CA, USA) for cDNA library preparation. cDNA libraries were selected for fragment sizes of between 120–210 bp. Sample cDNA libraries were quantified by Qubit, and the size distribution of libraries was characterized for each library using a Bioanalyzer 2100. The average library size for each sample was determined and normalized for stoichiometric balance. Libraries were pooled and sequenced en masse. Libraries were initially characterized on an Illumina MiSeq using a MiSeq Reagent Nano v2 kit (300 cycles), enabling final stoichiometric adjustment prior to initiating longer runs on an Illumina NextSeq 500. For whole transcriptome RNA-Seq, each library was run on four lanes using an Illumina NextSeq 500/550 High Output Kit v2.5 (150 Cycles). Up to ten sample libraries were run at a time on the NextSeq 500. Each sequencing run resulted in eight separate.fastq files per sample, containing two paired reads per lane. Reads were converted to. fastq format using Bcl2Fastq (Illumina) and demultiplexed. Forward and reverse reads from the four lanes were concatenated per sample to produce single forward and reverse read files.

Raw, unpaired Illumina sequencing read file were imported into Geneious Prime 2020.2.4 software (Biomatters Ltd., Auckland, New Zealand; https://www.geneious.com) for analysis. Reads were paired and then trimmed with the package BBDuk (version 38.84) to remove Illumina adapter sequences. Trimmed reads were aligned to the most current genome sequence for the *E. acervulina* Houghton reference strain [12] (annotated NCBI assembly EAH001)

using the Geneious RNA mapper. After mapping, various metrics of expression for each transcript were calculated, including raw mapped reads, raw mapped transcripts, and normalized expression metrics such as Reads Per Kilobase Million (RPKM), Fragments Per Kilobase Million (FPKM), and Transcripts Per Million (TPM). To facilitate analysis, data tables were exported from Geneious as.csv files and further analyzed in Microsoft Excel (Redmond, WA, USA) to calculate mean and standard deviation (SD) of raw reads, raw transcripts, and TPM for biological replicates. Data analysis and visualization was aided by Daniel's XL Toolbox addin for Excel, version 7.3.4, by Daniel Kraus, Würzburg, Germany ([www.xltoolbox.net](www.xltoolbox.net)). Correlation of RNA-Seq TPM among biological replicates was depicted as a heat map constructed in R using the packages ggplot2 and reshape2. Mean transcript diversity among replicates at each time point was estimated by subtracting from one the sum of the square of each gene's TPM. Bias in transcription (the excess contribution of especially abundant transcripts) was expressed as one minus the transcript diversity.

Bar graphs were produced in Microsoft Excel. A matrix of the log2 TPM for the 50 most variable genes was created in Microsoft Excel and imported into R. A filter was applied to eliminate any gene transcribed <0.01 TPM in one or more biological replicates. To visualize hierarchical clustering transcription among biological replicates, heatmap.2, pheatmap, and ComplexHeatmap packages in R were used to represent the matrix of Euclidean distances.

## Differential expression analysis

The DESeq2 package [13], as implemented in Geneious, calculated pairwise differential expression of time points in grouped biological replicates. This produced several estimates, including differential expression confidence, differential expression log2 ratio, and differential expression adjusted *p*-value. To facilitate analysis, data tables were exported from Geneious as. csv files and further analyzed in Microsoft Excel. We used thresholds of log2 fold change (FC) ±1.5 with adjusted *p*-value <0.05 to determine significantly differentially expressed genes (DEGs). Final volcano plots showing log2 FC vs.-log10 adjusted *p*-value for DEGs were constructed in the R package Enhanced Volcano.

## Homology searching and functional genomics analysis

We sought functional information on genes of interest using annotations available for the *E. acervulina* Houghton reference strain in NCBI (assembly EAH001) and ToxoDB v51 [14] ([www.toxodb.org](www.toxodb.org)). To elucidate additional functions of genes, we utilized the functional analysis module in OmicsBox 1.4.12 (BioBam Bioinformatics, Valencia, Spain). Our workflow incorporated Blast2GO (Basic Local Alignment Search Tool, Gene Ontology) methodology [15], identifying similarities to known and predicted proteins, scanning InterPro domains, and assigning GO terms to each of the sequences associated with proteins identified by NCBI BLASTP (through CloudBlast). BLASTP settings included a BLAST expectation value of $1 \times 10^{-3}$ against non-redundant protein databases filtered for Apicomplexa, 100 BLAST hits, Word Size = 3, and a low complexity filter. The rest of the workflow was run with default settings. Additionally, the EggNOG (evolutionary genealogy of genes: Non-supervised Orthologous Groups) [16] mapper (version 1.0.3 with EggNOG 5.0.0) inferred orthology relationships, gene evolutionary histories, and functional annotations to improve the Blast2GO sequence characterization. The KEGG (Kyoto Encyclopedia of Genes and Genomes) [17] pathway mapper enabled biochemical pathway enrichment analysis. Blast2GO results and lists of GO annotation terms and KEGG pathways were exported from OmicsBox and analyzed further in Microsoft Excel. Additional information for gene homologs and orthologs/paralogs was found by searching data sets in ToxoDB.

Genes annotated to encode *E. acervulina* transcription factors were identified and amino acid sequences were obtained from ToxoDB. The amino acid sequences were processed through the Blast2GO functional analysis workflow with the above settings using *E. acervulina* as a BLASTP taxonomy filter to identify GO information. To find hits in *C. cayetanensis*, the workflow was run using *C. cayetanensis* as a taxonomy filter. RNA-Seq data for these genes were extracted and paired with the Blast2GO results. To find homologs of *T. gondii* ApiAP2 transcription factors [18] in *E. acervulina*, amino acid sequences for *T. gondii* were first obtained from ToxoDB. The sequences were loaded into OmicsBox 1.4.12 and the above functional analysis workflow was performed using *E. acervulina* as a BLASTP taxonomy filter. The amino acid sequences for the *E. acervulina* homologs were downloaded from ToxoDB. To identify *E. acervulina* ApiAP2 homologs in *C. cayetanensis*, BLASTP of the *E. acervulina* ApiAP2 homolog amino acid sequences was performed with the above settings with a *C. cayetanensis* taxonomy filter. Blast2GO results were exported and paired with RNA-Seq data in Microsoft Excel.

### RT-qPCR

RNA-Seq estimates of gene expression were validated on a subsample of genes using semi-quantitative RT-qPCR. To do so, we synthesized cDNA from 10 ng of DNAse-treated RNA from 2–3 biological replicates per time point using the SuperScript™ IV First-Strand Synthesis System (Invitrogen). Individual RT-qPCR reactions were prepared with SsoAdvanced Universal SYBR Green Supermix (Bio-Rad, Hercules, CA.), 200–400 μM each primer (S1 Table), and 1–2 μl diluted cDNA in a total volume of 10 μL. Reactions were carried out on a CFX96 thermal cycler (Bio-Rad). Beta tubulin expression was used as a reference to normalize expression levels of other genes using the Pfaffl comparative quantification cycle (Cq) method [19]. All samples were run in triplicate. The abundance of mRNA in each biological replicate and time point was compared to that in unsporulated samples (T0). Gene expression was estimated for the reference and target genes after averaging Cq values for each biological replicate at each timepoint. Each experiment was performed three times and the mean of expression change was calculated for each gene and time point. Gene expression fold change was log2 transformed. The data was analyzed by CFX Software version 3.1 (Bio-Rad) and Microsoft Excel. All primers were designed using NCBI Primer-Blast [20] and synthesized by Integrated DNA Technologies (Coralville, IA, USA). Primer sequences are listed in S1 Table.

## Results

### RNA-Seq provides a global view of the *E. acervulina* transcriptome and identified differentially expressed genes during oocyst development

We achieved nearly two billion 75-bp paired-end reads from RNA-Seq (19 samples, representing 2–3 biological replicates per time point during the 24-hour *E. acervulina* oocyst sporulation time course). Table 1 summarizes the RNA-Seq data. The number of reads per replicate and time point ranged from 44,335,874 (T20-C) to 195,666,302 (T12-B) (mean = 103,669,059 ± 37,595,932). A high percentage of reads (92.2%) mapped to the *E. acervulina* genome (range = 84.4–95.0%). The highest percentage of reads that mapped was for sample T12-C. Importantly, each biological replicate yielded at least 40 million mapped reads per time point, sufficient coverage to provide a global view of transcription [21].

We used Transcripts Per Million (TPM) as a normalization method to summarize expression of the 6,867 annotated genes. S2 Table compiles TPM for biological replicates and time points. Strong correlation in expression characterized biological replicates at each time point

**Table 1. *E. acervulina* oocyst RNA-Seq reads depth and quality.**

| Time | Replicate* | | Reads | % mapped |
|---|---|---|---|---|
| 0 | A | | 153,222,282 | 92.4 |
| | B | | 78,702,846 | 91.5 |
| | | C | 93,262,116 | 91.7 |
| 4 | A | | 93,014,892 | 92.1 |
| | B | | 94,418,254 | 84.4 |
| | | C | 145,426,320 | 91.7 |
| 8 | A | | 89,215,090 | 91.1 |
| | B | | 107,410,556 | 93 |
| | | C | 109,236,344 | 92.5 |
| 12 | A | | 100,428,700 | 92.2 |
| | B | | 195,666,302 | 92.8 |
| | | C | 161,356,814 | 95 |
| 16 | B | | 78,783,020 | 94.8 |
| | | C | 62,231,814 | 94.6 |
| 20 | B | | 71,554,584 | 92.6 |
| | | C | 44,335,874 | 93 |
| 24 | A | | 114,688,324 | 92.8 |
| | B | | 68,638,488 | 92.2 |
| | | C | 108,119,510 | 91.4 |
| | Total | | 1,969,712,130 | 92.2 |
| | Mean | | 103,669,059 | 92.2 |
| | Median | | 94,418,254 | 92.4 |
| | SD | | 37,595,932 | 2.2 |

*RNA-Seq was performed on 2–3 replicates (designated A, B, and C) initially (T0) and every four hours thereafter (T4-T24).

(Fig 1); the least correlation among replicates ($r = 0.768$) occurred at 12 hours (T12-C). Expression at hours 8–16 correlated least with other time points, particularly at earlier stages, indicating that parasites undergo especially marked changes in gene expression during this time interval. Therefore, hours 8–16 represent an important developmental transition point during sporulation.

## Genes expressed at high levels throughout sporulation

To identify highly expressed genes that might serve as markers of oocyst viability, we first sought to understand how uniformly genes contribute to the transcriptome during oocyst development. We examined mean TPM of biological replicates at each time point (S3 Table). A large majority of the ~6,900 annotated genes contributed few transcripts (mean TPM = 145.62, < .02% of total transcripts). Only ~600–900 genes (<15% of the genes) at each time point were expressed >100 TPM (Table 2). To better understand global expression, we conducted an analysis of transcriptional bias at each time point. That bias reflects the variance in how much each transcript contributes to the total. Transcription was slightly more biased in genes expressed at TPM >100 (Fig 2). Among all genes (and those >100 TPM), the greatest transcriptional bias was observed at 4 hours. However, bias did not increase more than 2.5% for any time point. The amount of bias observed for all genes and those with TPM >100 correlated well ($r = 0.999$).

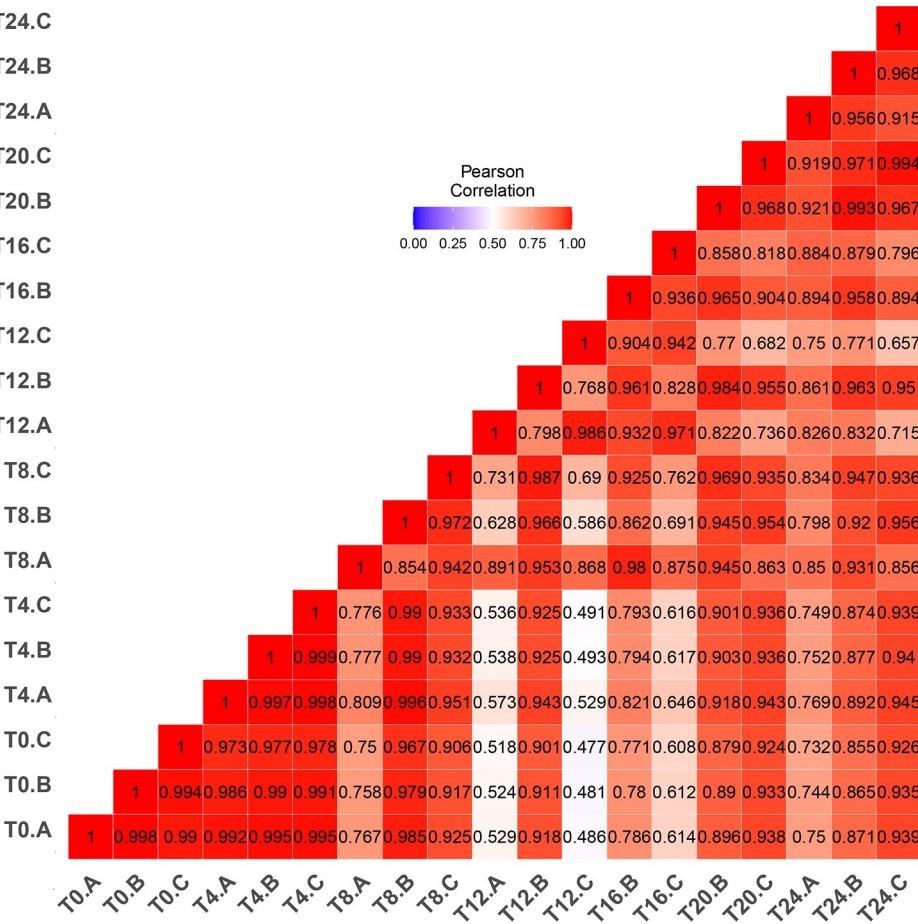

**Fig 1. Correlation of global transcription among biological replicates (A-C) for ~6,900 *E. acervulina* genes as oocysts sporulate.** Transcripts Per Million (TPM) were compared for each biological replicate every four hours during sporulation. Individual squares represent Pearson Product correlations among replicates at each time point. Correlations were greatest initially (hours 0–4 [T0-T4]) and after sporulation's completion (hours 20–24: [T20-T24]).

Maximum expression for any gene ranged from 30,654 (T20) to 52,966 (T4) (accounting for 3.1–5.3% of total transcripts, Table 2). To narrow our search for important and easily amplified biomarkers, we thereafter established 1,000 TPM (0.1% of the total) as a cutoff for

**Table 2. RNA-Seq expression analytics for *E. acervulina* oocysts during sporulation.**

| Time (hours) | Most transcribed gene (TPM) | Number of genes | | | | |
| --- | --- | --- | --- | --- | --- | --- |
| | | TPM >1000 | TPM >100 | TPM <100 | Up-regulated* | Down-regulated* |
| 0 | 42,934 | 95 | 826 | 6,041 | N/A | N/A |
| 4 | 52,966 | 78 | 605 | 6,262 | 794 | 780 |
| 8 | 41,238 | 131 | 751 | 6,116 | 1,554 | 943 |
| 12 | 43,379 | 142 | 853 | 6,014 | 1,897 | 836 |
| 16 | 34,457 | 141 | 923 | 5,944 | 1,893 | 881 |
| 20 | 30,654 | 131 | 826 | 6,041 | 1,663 | 796 |
| 24 | 30,802 | 127 | 836 | 6,031 | 1,843 | 1,085 |

Transcripts Per Million (TPM), averaged among biological replicates, for each time point.

*Genes expressed significantly more, or less, than initially (T0); pairwise differential expression ±1.5 fold change (FC) and adjusted $p$-value <0.05.

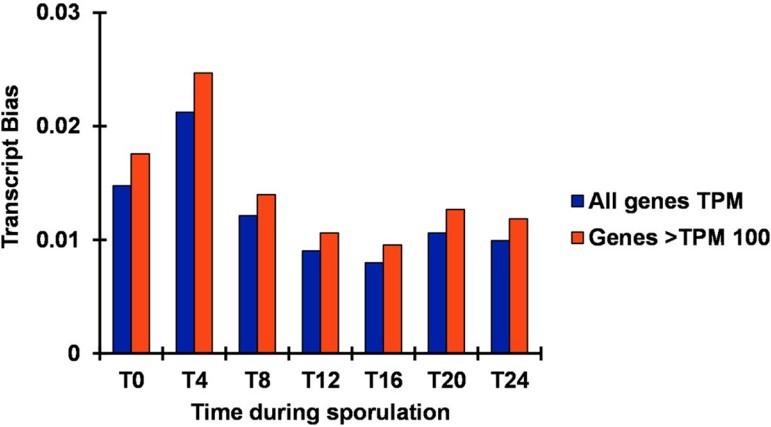

**Fig 2. Transcript bias during oocyst sporulation.** The bias in transcription (when a subset of genes most disproportionately contribute to transcripts) is greatest earlier in sporulation and peaks at hour 4 (T4). Parallel temporal patterns in transcriptional bias hold whether the analysis is performed on all genes (blue) or restricted to those each contributing at least 100 transcripts per million (TPM >100) (orange). The mean among the biological replicates is shown for each timepoint.

expression, reasoning that those genes comprising an appreciable portion of the total transcripts throughout sporulation would make for especially productive diagnostic targets.

Seeking markers of viable and infectious oocysts, we examined which transcriptional activities occur throughout the entire sporulation time course. At any given time, as few as 78 or as many as 142 genes were transcribed >1,000 TPM (at 4 and 12 hours, respectively; Table 2). Of these, 53 genes were transcribed > 1,000 TPM throughout the 24-hour time course (Fig 3). For present purposes, we designate as "constitutive" the expression of these genes which were highly expressed throughout the 24-hour period we studied. We do not, however, know which of these genes are transcribed throughout the entire life cycle, including those stages infecting host cells. Of these, 49% (26/53) are annotated as hypothetical proteins in current NCBI and ToxoDB genome annotations; among these are some of the most highly-expressed genes. Expression of a subset of genes peaked at T4, diminished at T12, peaked again at T20, and again decreased slightly by T24.

**Inferred function of highly expressed genes.** Notably high expression characterized a few genes. These serve as the most promising candidate biomarkers for viability: expression of some exceeded a mean of 25,000 TPM over the entire time course (Fig 3, S4 Table). Two of these genes (*EAH_00004100* and *EAH_00004110*) are located near each other in the genome and are functionally related (see below). *EAH_00004110* had the highest consistent TPM across all time points (overall mean = 33,808). Other highly-expressed genes included *EAH_00037050* (encoding a hypothetical protein), *EAH_00033530* (encoding an oocyst wall protein), *EAH_00034270* (encoding a zinc finger DHHC domain-containing protein), and *EAH_00004780* (encoding a hypothetical protein). On the lower end of the expression spectrum (~ ≤2,100 TPM), several genes define general and housekeeping functions (such as beta tubulin, aquaporin, and glycerol-3-phosphate dehydrogenase).

Because the nominal function of about half these genes cannot be inferred from existing genome annotations, we pursued additional means to explore gene function. We used Blast2GO to search gene homologs and identify function of proteins. After performing a BLASTP search narrowed to Apicomplexa, 21 of the 26 genes initially identified only as encoding "hypothetical proteins" yielded a more informative protein hit in the results returned (bolded in S4 Table). The protein sequences encoded by five genes (italicized in S4 Table) either

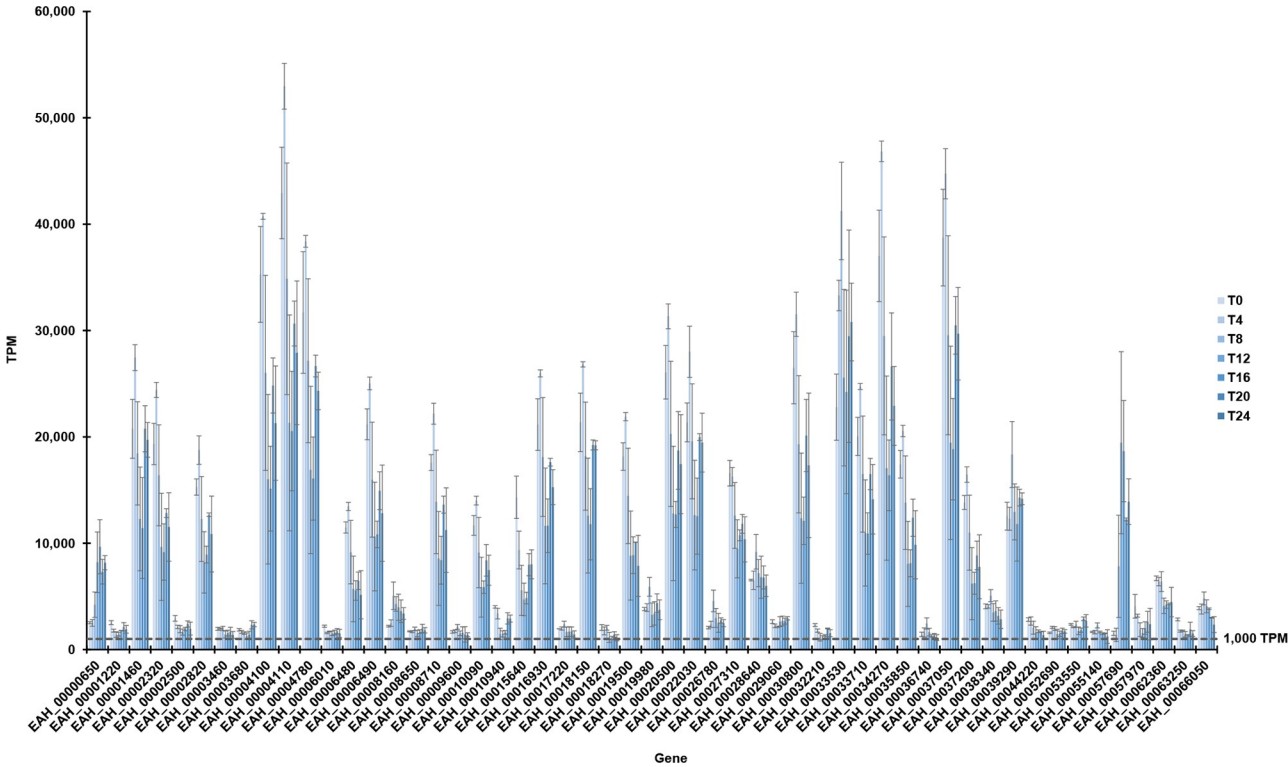

**Fig 3. Dynamic constitutive expression of major transcripts throughout _E. acervulina_ sporulation (T0-T24).** Fifty-three genes contributed at least 1000 transcripts per million (0.1%) throughout the 24-hour period. Some (_i.e._ EAH_0000120; EAH_00002500) contributed a modest and relatively unchanging proportion of total transcripts, whereas others (_i.e._ EAH_00004100 and EAH_00004110) contributed more to the total and underwent greater change through time. (Mean TPM and SD among biological replicates).

matched to a hypothetical or uncharacterized protein in another species or did not have any hit (_EAH_00055140_).

Importantly, all but four genes (_EAH_00019980_, _EAH_0053550_, _EAH_0055140_, _EAH_0057970_) matched proteins in _C. cayetanensis_. Of these 49 genes, 23 matched the same annotated protein in _Eimeria_ or a related species, 5 matched another annotated protein, and 21 matched to an uncharacterized or hypothetical protein in another species. With a few exceptions, the uncharacterized protein hits matched to original hypothetical protein annotations in _E. acervulina_. For the highly-expressed genes described above, only _EAH_00033530_ (oocyst wall protein) and _EAH_00034270_ (palmitoyltransferase ZDHHC15-like) had informative hits in _C. cayetanensis_.

Of the 53 total genes, 45 (85%) were successfully annotated with at least one Gene Ontology (GO) term. The genes without GO terms were mostly those encoding hypothetical proteins. In total, there were 285 GO terms annotated (S4 and S5 Tables), categorized into subontologies of Biological Process (BP, 109), Molecular Function (MF, 92), and Cellular Component (CC, 84). The most dominant BP terms included those related to protein ubiquitination, proteolysis, translation, and those encompassing microtubule/actin/cytoskeleton, response to drug, transmembrane transport, nitrogen compound metabolism, and mitosis. In addition, many terms pertained to regulation and cellular responses. The most numerous CC terms included integral component of membrane, cytoplasm, cytosol, nucleus, and ubiquitin ligase complex. In addition, other terms were identified that we could group into cytoskeleton and the

endomembrane system. The dominant MF terms were ATP binding, protein, cytoskeleton, and ribosome functions, fatty acid synthesis enzymes, and ubiquitin ligase binding. Other terms pertained to protein modification and calcium binding/activity.

We chose to focus on some of the highly-expressed genes for further functional analysis. *EAH_00004110*, annotated only to encode a hypothetical protein, matched to *EAH_00004100* by BLAST and other Cation-transporting ATPases in related taxa (it was more like *E. maxima* Cation-transporting ATPase than EAH_00004100, however). These genes were enriched for CC GO terms, including mitochondrion and integral membrane component (S4 Table). For some of the higher-expressed genes that encode hypothetical proteins (see above), Blast2GO revealed more information. *EAH_00037050* encodes a protein related to membrane proteins in *Neospora caninum* and *T. gondii* and *EAH_00004780* encodes a protein related to a putative transmembrane protein in *T. gondii*. For each of these genes, a single GO term was identified (integral component of membrane). *EAH_00033530*, the third highest expressed gene (by mean TPM) had several GO terms, including chitin binding, extracellular region, and integral membrane component. *EAH_00034270* also had several GO terms grouped into multiple sub-ontologies, including integral component of membrane, protein-cysteine S-palmitoyltransfer-ase activity (protein S-acyltransferase), nitrogen compound metabolism, macromolecule modification; and primary and cellular metabolic processes. As a point of comparison, several genes encoding general function proteins had a relatively high number of terms (*e.g.* *EAH_00000650*, *EAH_00029060*, *EAH_00052690*) and they were toward the middle to low end of the spectrum in terms of mean TPM across sporulation.

We also applied KEGG pathway analysis to constitutively expressed genes, assigning gene-encoding enzymes to large metabolic networks. Of the 53 genes, 13 were identified as involved in more than one metabolic pathway (Fig 4A and shaded in S4 Table). Interestingly, none of these genes included any of the higher-expressed genes noted above. In fact, these genes were among the lowest-expressed throughout the time course: in the range of ~1,600–13,000 mean TPM (Fig 3, S4 Table). We also noted that some of these genes were among those with a larger number of GO annotations than others. A total of 18 pathways were identified for these genes. The major pathways defining these genes included thiamine metabolism (18.5%), purine metabolism (14.8%), glycerophospholipid metabolism (7.4%), fatty acid elongation (7.4%), and biosynthesis of unsaturated fatty acids (7.4%). Of the genes linked to a pathway (nicotinate and nicotinamide metabolism), *EAH_00019500* had the highest expression (mean TPM = 13,241, but highest expression at T0-T4). The modestly and constitutively expressed genes encode enzymes in metabolic pathways that underlie basic housekeeping functions.

## Expressed transcription factors

As part of our RNA-Seq analysis, we wished to investigate expression of transcription factors (TFs) and if they may play a factor in sporulation. We first searched for *E. acervulina* genes annotated to encode TFs, identifying 17 genes (S6 Table). Only one of these (*EAH_00013480*, encoding the CCR4-NOT transcription complex subunit) was expressed over 100 TPM at any time during sporulation. Seven putative TFs were most expressed at T24 and five of them met our criteria for increased expression (>1.5 log2 fold change and *p*<0.05). The most differen-tially expressed gene in T24 oocysts was *EAH_00043880* (encoding transcription elongation factor TFIIS) although TPM did not exceed 2.9 during the entire time course GO terms for these genes mostly related to transcription and translation regulation. Blast2GO found homo-logs of these TFs in *C. cayetanensis* and the top hits generally had similar annotations.

We also sought to find homologs of ApiAP2 TFs, since these are recognized as the key fac-tors regulating apicomplexan stage differentiation during development [18, 22]. We focused

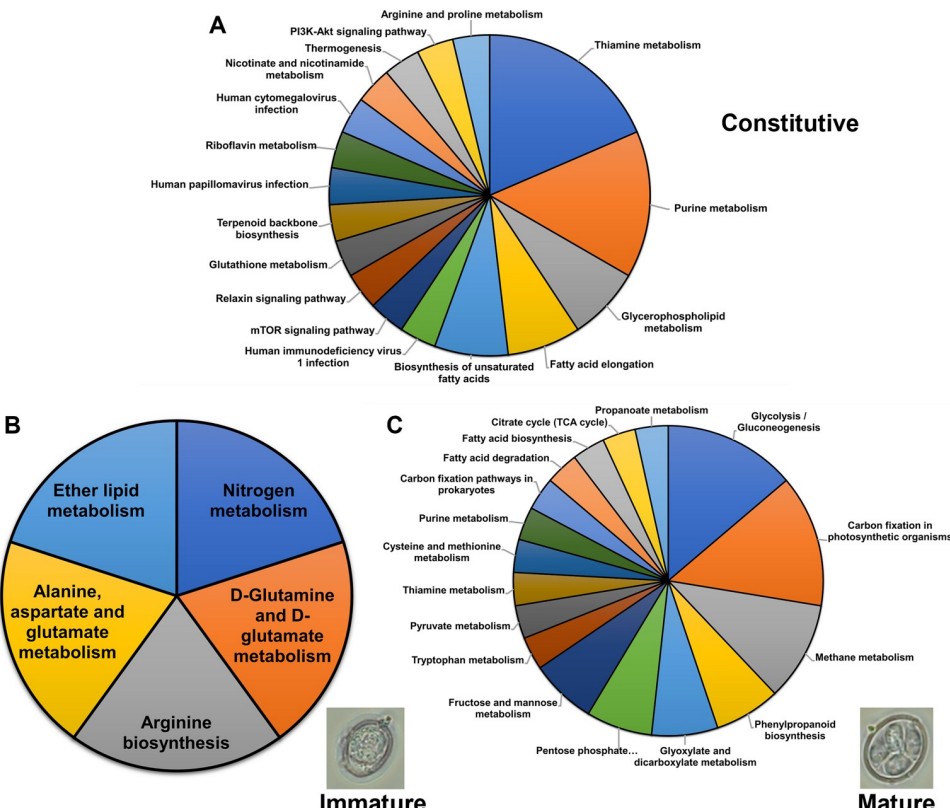

**Fig 4. KEGG Biochemical pathway assignment for constitutively expressed genes and differentially expressed genes in immature and mature oocysts.** (A) Constitutively expressed genes (13 genes were involved in 17 pathways). (B) Immature oocysts, T0 (2 genes, 5 pathways). (C) Mature oocysts, T24 (9 genes, 18 pathways). Representative photomicrographs of immature and mature oocysts are shown.

on ApiAP2 TFs in *T. gondii* [18] and Blast2GO provided functional characterization of these genes and identified homologs in *E. acervulina* (S7 Table). Of the 67 *T. gondii* genes, all but 7 had a hit in *E. acervulina*. An analysis of the BLAST results found that matching hits were mostly hypothetical proteins. Also, multiple *T. gondii* genes had the same best hit in *E. acervulina*. Therefore, we identified 49 unique *E. acervulina* ApiAP2 homologs. Again, GO annotations for these *E. acervulina* hits pertained to transcriptional regulation. Eight of the 61 hits did not have any associated GO annotations. No KEGG pathways were identified for the *E. acervulina* matches. We blasted these 49 unique *E. acervulina* ApiAP2 homolog amino acid sequences against *C. cayetanensis*, looking for hits already annotated as ApiAP2 TFs (listed in S7 Table). Of the 49 sequences, 31 (63%) identified a sequence annotated as an ApiAP2 family member. Other TF hits were sometimes returned as well.

From the ApiAP2 homolog data set, two genes had mean TPM >100 over the entire time course. Expression of these genes did not significantly differ in immature and mature oocysts. Twenty-two unique gene homologs in *E. acervulina* met our criteria for significant differential expression (bolded in S7 Table). However, five of these genes (*EAH_00000660*, *EAH_00000670*, *EAH_00021020*, *EAH_00046540*, and *EAH_00051470*) were expressed >37-fold (log2 = 5.22) in mature vs. immature oocysts. Some of these genes (although not abundant) were markedly increased for expression over the time course. Therefore, these TFs may help orchestrate sporulation.

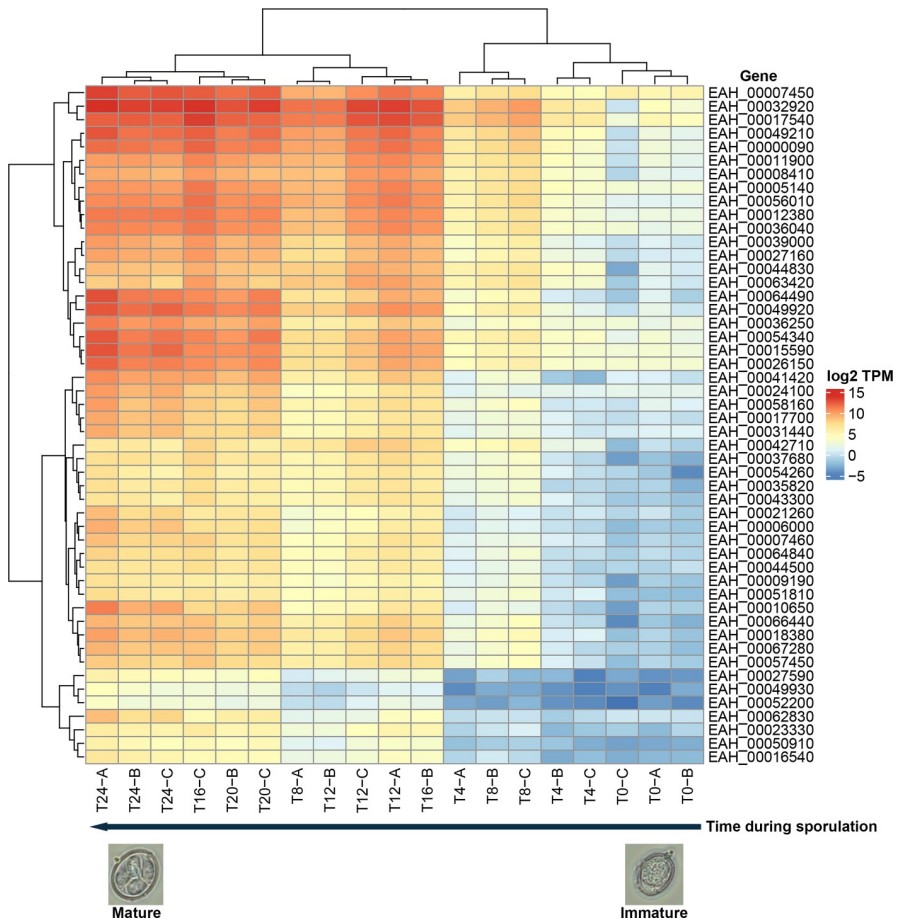

**Fig 5. Coordinated change in the top 50 genes undergoing the greatest extent of variation in expression through time.** Maximal expression of such genes occurs in mature oocysts (left side of heatmap). Note considerable agreement among biological replicates. Expression is displayed as log2 TPM.

## Temporal dynamic gene expression defines temporally variable patterns and markers for immature and mature oocysts

We sought maturation biomarkers by identifying genes whose stage-specific differential expression identified immature, maturing or fully mature oocyst cohorts. Mature oocyst cohorts are enriched for fully sporulated oocysts. To conduct a preliminary, global analysis, we analyzed expression variance for all biological replicates through time. A heatmap of the 50 most temporally-variable genes revealed consistent patterns in differential gene expression over the first 24 hours of sporulation (Fig 5). A graded, chronological increase in expression is evident for many genes. Of these genes, upregulation for some began at T8, and almost all increased expression by T12; some reached highest expression at T24. A group of down-regulated genes defines immature oocysts (not strongly expressed after hour 8). The greatest consistency among biological replicates occurred initially (T0-T4) and in mature oocysts (T24). Thus, many genes may serve as biomarkers for sporulated, infectious oocysts of *E. acervulina*.

## Genes especially expressed in immature oocysts

Comparing expression at hour 0 and 24 identified genes most associated with either immature or mature oocysts. Biological replicates at each timepoint were pooled for this purpose.

Initially, we considered all genes whose expression differed at hours 0 and 24 (S8 Table); we then examined such genes meeting additional thresholds. Of the ~6900 genes, 2,928 were significantly differentially expressed (1,843 up-regulated, 1,085 down-regulated) using +/- 1.5 log2 fold change (FC) and adjusted $p < 0.05$ as threshold criteria (Fig 6, Table 2).

Among these differentially-expressed genes, we then focused on those expressed with >1,000 TPM at T0. A group of only eight genes met this expression threshold (Fig 7). Notably, each of these genes constituted a greater proportion of total transcripts at T0 than at any other time point. Table 3 shows the mean TPM for each of these genes throughout sporulation and log2 FC compared to T24 by differential expression analysis. Initial TPM of these genes exceeded the next-highest time point by 1.6–10.4-fold (for *EAH_00002160* and *EAH_00006770*, respectively). Although initial expression of these genes (T0) significantly exceeded that of mature oocysts (T24), differential expression never exceeded 12-fold (*EAH_00006770*, log2 = 3.6). All genes were expressed more in at least one intermediate stage compared to T24, and five experienced a secondary peak at T20 before decreasing again at T24. *EAH_00006770* was expressed >10,000 TPM at hour zero, a proportion at least twice as great as any other gene in the group. The next highest expressed gene in the group (*EAH_00050370*) had 5-fold (log2 = 2.3) higher expression than at T24. *EAH_00048170* (the third highest expressed gene) was expressed ~3.8-fold (log2 = 1.94) more than at T24.

**Functions of genes expressed most in immature oocysts.** Blast2GO analysis found that of the four genes originally only annotated as encoding a hypothetical protein, two (*EAH_00000360*, *EAH_00034940*) matched to a more informative sequence (bolded in S9 Table). The top two up-regulated genes at T0, *EAH_00006770* and *EAH_00050370* (italicized in S9 Table), returned matches only to hypothetical or uncharacterized proteins from related species. A hit was identified for *C. cayetanensis* for each of the eight genes. Three of the *C. cayetanensis* hits were annotated only as hypothetical or uncharacterized proteins, but three others matched to a protein with similar annotation in another species. Two other hits matched to different proteins, including an endoribonuclease YBEY, chloroplastic and alkyldihydroxyacetonephosphate synthase, peroxisomal (the latter being the hit for *EAH_00034940*).

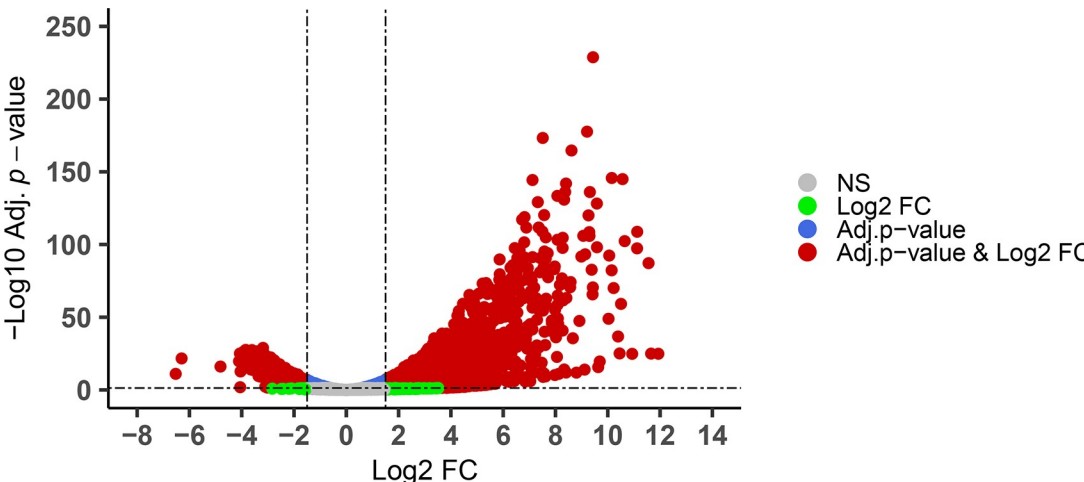

**Fig 6. Differentially expressed genes in mature and immature oocysts of *E. acervulina*.** Expression at hour 24 was compared to expression at hour 0, using ±1.5 log2 Fold Change (FC) with an adjusted $p < 0.05$ as threshold criteria. By these criteria, 2,928 significantly differentially expressed genes were identified (1,843 up-regulated, 1,085 down-regulated). NS indicates non-significant.

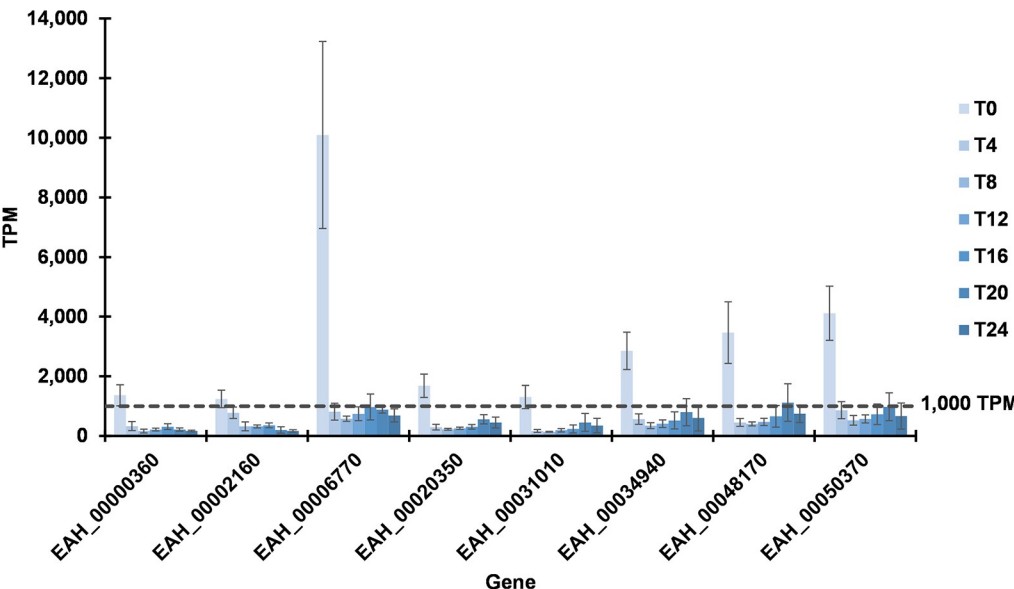

**Fig 7. Significantly up-regulated major transcripts in immature oocysts.** Eight genes contributed at least 1,000 transcripts per million (TPM) and were transcribed more at hour 0 than at hour 24 (using >1.5 log2 fold change and adjusted $p$ <0.05 as threshold criteria).

Of these same 8 genes, all except *EAH_00006770* were annotated with at least one Gene Ontology (GO) term and some terms were assigned to genes originally annotated as hypothetical proteins. In total, there were 37 GO terms annotated (S9 and S10 Tables) categorized into subontologies of BP (11), MF (10), and CC (16). Three genes were associated with only one GO term (integral component of membrane [*EAH_00000360*, *EAH_00050370*] or hydrolase activity [*EAH_00020350*]). Three GO terms were associated with *EAH_00048170*, including proteolysis, serine-type endopeptidase activity, and membrane. Most of the terms overall were assigned to the remaining genes (*EAH_00002160*, *EAH_00031010*, and *EAH_00034940*) and encompassed a diversity of important functions and processes. *EAH_00002160* (encodes a microtubial-binding protein) had 15 associated GO terms including autophagy, organelles (apicoplast, Golgi apparatus), SNARE binding, and intracellular transport. *EAH_00034940* (encodes a hypothetical protein) matched to lactate dehydrogenase [LDH] in *E. necatrix*) and had 10 GO terms, including mitochondria and peroxisome components and processes, oxidoreductase activity, Flavin Adenine Dinucleotide (FAD) binding, and ether lipid biosynthesis. *EAH_00031010* (encodes Nicotinamide adenine dinucleotide phosphate [NADP]-specific glutamate dehydrogenase) also had several GO term annotations, including those involved in glutamate metabolism, nucleic acid metabolic process/nucleotide binding, apicoplast, and cytosol. KEGG pathway analysis performed on this gene set found two genes (*EAH_00031010* and *EAH_00034940*, shaded in S9 Table) were associated with 5 pathways (Fig 4B). These pathways included nitrogen metabolism, ether lipid metabolism, and several amino acid (glutamine, glutamate, arginine, alanine, aspartate) metabolic pathways. *EAH_00048170* (while not identified as integral to a metabolic pathway), encodes a putative subtilase family serine protease that appears to have broad and diverse activity. Genes encoding proteins linked to these pathways, although not with highest TPM, were still ~3-fold higher expressed than at any other time point during sporulation. This indicates these pathways may be unique to immature oocysts.

**Table 3. Significantly differentially expressed genes between immature (T0) and mature (T24) *E. acervulina* oocysts.**

**T0 Up-regulated genes**

| Gene | T0 TPM | T12 TPM | T24 TPM | Mean TPM* | Log2 FC** | Adjusted p-value | Original annotation |
|---|---|---|---|---|---|---|---|
| EAH_00006770 | 10,092 | 739 | 685 | 2,232 | 3.59 | 4.63E-21 | hypothetical protein, conserved CDS |
| EAH_00050370 | 4,112 | 569 | 662 | 1,241 | 2.32 | 3.21E-06 | hypothetical protein, conserved CDS |
| EAH_00048170 | 3,465 | 466 | 744 | 1,058 | 1.94 | 3.53E-06 | subtilase family serine protease, putative CDS |
| EAH_00034940 | 2,853 | 407 | 601 | 891 | 1.94 | 1.46E-04 | hypothetical protein CDS |
| EAH_00020350 | 1,678 | 253 | 446 | 547 | 1.64 | 6.09E-05 | haloacid dehalogenase-like hydrolase domain-containing protein, putative CDS |
| EAH_00000360 | 1,363 | 212 | 164 | 407 | 2.81 | 1.24E-16 | hypothetical protein, conserved CDS |
| EAH_00031010 | 1,305 | 188 | 346 | 410 | 1.61 | 1.37E-03 | NADP-specific glutamate dehydrogenase, putative CDS |
| EAH_00002160 | 1,238 | 318 | 163 | 502 | 2.68 | 4.59E-14 | microtubial-binding protein, putative CDS |

**T24 up-regulated genes**

| Gene | T0 TPM | T12 TPM | T24 TPM | Mean TPM* | Log2 FC** | Adjusted p-value | Original annotation |
|---|---|---|---|---|---|---|---|
| EAH_00011300 | 1,622 | 8,200 | 22,929 | 8,368 | 3.9 | 4.17E-31 | hypothetical protein, conserved CDS |
| EAH_00059200 | 588 | 43,379 | 19,815 | 19,577 | 5.1 | 1.79E-19 | SAG family member CDS |
| EAH_00059950 | 448 | 30,353 | 14,306 | 14,459 | 5.0 | 1.67E-17 | SAG family member CDS |
| EAH_00057690 | 1,531 | 19,450 | 13,911 | 10,216 | 3.4 | 2.97E-36 | profilin family protein, putative CDS |
| EAH_00032920 | 9.0 | 7,138 | 11,461 | 5,134 | 10.0 | 1.17E-49 | hypothetical protein CDS |
| EAH_00045000 | 347 | 7,074 | 9,793 | 4,856 | 4.9 | 1.08E-55 | actin depolymerizing factor, putative CDS |
| EAH_00000650 | 2,524 | 8,211 | 8,164 | 5,836 | 1.9 | 1.55E-10 | actin, putative CDS |
| EAH_00007450 | 39.7 | 1,624 | 6,839 | 2,177 | 7.6 | 5.61E-121 | glyceraldehyde-3-phosphate dehydrogenase, putative CDS |
| EAH_00022290 | 50.3 | 656 | 5,848 | 1,502 | 7.0 | 8.73E-78 | hypothetical protein CDS |
| EAH_00017770 | 105 | 4,878 | 5,704 | 3,328 | 5.8 | 2.23E-34 | hypothetical protein, conserved CDS |
| EAH_00013230 | 672 | 5,962 | 5,156 | 3,607 | 3.1 | 1.11E-22 | hypothetical protein, conserved CDS |
| EAH_00049200 | 125 | 10,834 | 4,768 | 5,013 | 5.2 | 2.12E-16 | hypothetical protein CDS |
| EAH_00017540 | 17.9 | 5,311 | 4,614 | 3,031 | 8.0 | 9.87E-56 | hypothetical protein, conserved CDS |
| EAH_00049920 | 1.8 | 761 | 4,414 | 1,224 | 11.1 | 4.46E-98 | hypothetical protein, conserved CDS |
| EAH_00002690 | 44.8 | 857 | 4,400 | 1,286 | 6.7 | 1.38E-85 | serine protease inhibitor, putative CDS |
| EAH_00015590 | 7.0 | 478 | 4,277 | 1,050 | 9.3 | 8.62E-137 | hypothetical protein, conserved CDS |
| EAH_00049210 | 3.0 | 2,433 | 4,033 | 1,737 | 10.2 | 8.80E-71 | hypothetical protein, conserved CDS |
| EAH_00044990 | 169 | 3,263 | 3,951 | 2,136 | 4.7 | 4.61E-66 | Hsp20/alpha crystallin domain-containing protein, putative CDS |
| EAH_00058140 | 43.3 | 5,237 | 3,851 | 2,757 | 6.4 | 1.57E-31 | hypothetical protein, conserved CDS |
| EAH_00064490 | 1.0 | 338 | 3,821 | 941 | 10.9 | 1.53E-25 | hypothetical protein CDS |
| EAH_00040110 | 62.0 | 7,430 | 3,598 | 3,449 | 5.8 | 3.22E-25 | hypothetical protein, conserved CDS |
| EAH_00054340 | 5.1 | 459 | 3,589 | 984 | 9.4 | 2.23E-83 | lactate dehydrogenase, putative CDS |
| EAH_00047270 | 18.9 | 598 | 2,878 | 902 | 7.4 | 2.04E-112 | peroxiredoxin, putative CDS |
| EAH_00001100 | 84.6 | 10,150 | 2,853 | 4,024 | 5.0 | 1.47E-17 | hypothetical protein, conserved CDS |
| EAH_00026150 | 8.8 | 406 | 2,748 | 763 | 8.4 | 6.79E-137 | hypothetical protein, conserved CDS |
| EAH_00000090 | 3.7 | 1,938 | 2,716 | 1,338 | 9.4 | 1.99E-66 | Microneme protein etmic-2/7h, related CDS |
| EAH_00031210 | 116 | 9,179 | 2,433 | 3,890 | 4.4 | 4.75E-14 | hypothetical protein, conserved CDS |
| EAH_00034850 | 107 | 9,976 | 2,390 | 3,827 | 4.5 | 1.04E-19 | hypothetical protein, conserved CDS |
| EAH_00012380 | 4.2 | 1,120 | 2,336 | 976 | 9.2 | 2.55E-178 | hypothetical protein, conserved CDS |
| EAH_00001210 | 27.7 | 4,277 | 2,213 | 2,075 | 6.3 | 1.61E-34 | serine proteinase inhibitor TgPI-2, putative CDS |
| EAH_00011540 | 527 | 2,473 | 2,143 | 1,542 | 2.2 | 1.90E-16 | 14-3-3 protein, putative CDS |
| EAH_00049190 | 54.1 | 7,835 | 2,037 | 3,116 | 5.2 | 1.76E-18 | hypothetical protein CDS |
| EAH_00053040 | 19.0 | 571 | 1,911 | 681 | 6.8 | 3.17E-102 | superoxide dismutase, putative CDS |
| EAH_00011680 | 96.5 | 8,727 | 1,903 | 3,305 | 4.3 | 1.07E-13 | SAG family member CDS |

*(Continued)*

**Table 3.** (Continued)

| | | | | | | |
|---|---|---|---|---|---|---|
| EAH_00043500 | 112 | 869 | 1,739 | 728 | 4.1 | 1.62E-39 | hypothetical protein, conserved CDS |
| EAH_00036040 | 5.0 | 1,048 | 1,731 | 808 | 8.6 | 2.27E-165 | hypothetical protein, conserved CDS |
| EAH_00062880 | 124 | 616 | 1,719 | 674 | 3.9 | 4.02E-33 | uridine phosphorylase, putative CDS |
| EAH_00019100 | 70.9 | 5,744 | 1,699 | 2,388 | 4.6 | 1.35E-17 | myosin light chain TgMLC1, putative CDS |
| EAH_00036250 | 5.4 | 164 | 1,586 | 404 | 8.3 | 1.44E-131 | hypothetical protein, conserved CDS |
| EAH_00039520 | 123 | 1,300 | 1,568 | 836 | 3.8 | 1.86E-39 | hypothetical protein, conserved CDS |
| EAH_00056010 | 2.4 | 1,445 | 1,544 | 910 | 9.4 | 1.69E-229 | hypothetical protein, conserved CDS |
| EAH_00057900 | 13.9 | 458 | 1,507 | 545 | 6.9 | 2.29E-112 | Fructose-bisphosphate aldolase, related CDS |
| EAH_00009280 | 16.1 | 2,311 | 1,411 | 1,117 | 6.4 | 1.60E-34 | hypothetical protein CDS |
| EAH_00017580 | 32.2 | 3,021 | 1,377 | 1,353 | 5.4 | 2.43E-26 | hypothetical protein, conserved CDS |
| EAH_00002760 | 74.5 | 5,292 | 1,373 | 2,107 | 4.2 | 2.52E-17 | hypothetical protein, conserved CDS |
| EAH_00067850 | 12.6 | 431 | 1,290 | 484 | 6.8 | 1.46E-119 | Fructose-bisphosphate aldolase, related CDS |
| EAH_00054350 | 30.1 | 2,849 | 1,214 | 1,299 | 5.3 | 1.83E-19 | hypothetical protein, conserved CDS |
| EAH_00010660 | 14.7 | 95.7 | 1,182 | 289 | 6.4 | 7.91E-59 | hypothetical protein, conserved CDS |
| EAH_00010650 | 0.3 | 49.8 | 1,175 | 238 | 11.6 | 6.69E-88 | hypothetical protein, conserved CDS |
| EAH_00065220 | 13.8 | 1803 | 1,138 | 910 | 6.3 | 2.01E-28 | hypothetical protein CDS |
| EAH_00022310 | 12.4 | 435 | 1,133 | 417 | 6.6 | 1.02E-96 | hypothetical protein, conserved CDS |
| EAH_00005140 | 6.9 | 1251 | 1,126 | 724 | 7.5 | 5.36E-174 | hypothetical protein, conserved CDS |
| EAH_00002610 | 45.5 | 542 | 1,092 | 474 | 4.7 | 2.76E-31 | Heat shock protein, related CDS |
| EAH_00037890 | 27.8 | 3146 | 1,070 | 1,359 | 5.2 | 5.88E-19 | hypothetical protein, conserved CDS |
| EAH_00044180 | 47.2 | 3601 | 1,066 | 1,477 | 4.6 | 6.05E-25 | hypothetical protein, conserved CDS |
| EAH_00041420 | 1.4 | 170 | 1,060 | 313 | 9.6 | 6.96E-129 | peroxisomal catalase, putative CDS |

Transcripts Per Million (TPM), averaged among biological replicates, for time points T0, T12, and T24. Genes are sorted by TPM at T0 or T24.

*Mean TPM = mean gene TPM for all biological replicates and time points

**Genes expressed significantly more in T0-T24 pairwise comparison, meeting thresholds of >1.5 log2 fold change (FC) and adjusted $p$ <0.05 and had >1,000 TPM assessed at T0 and T24. Log2 FC and adjusted $p$-value were determined from DeSeq2

## Genes especially expressed in mature oocysts

Finally, we identified 56 genes expressed >1,000 TPM at T24 and to a significantly greater extent compared to T0. As illustrated in Fig 8, some genes increased in relative abundance as early as 8 or 12 hours after commencing sporulation, some accounting for their maximum proportion of total transcripts midway through the study (at hour 12 or 16). Notably, two genes encoding surface antigens (SAGs) (which had the highest overall mean TPM throughout the time course) increased ~2-fold from T8 to T12. Of the 56 genes, 52% (n = 29) peak at an intermediate time point, and not at T24. Approximately 45% (n = 25) of the genes were expressed higher at T12 than T24 (Table 3). Strong expression of these genes indicates that the oocysts are undergoing, or have successfully completed, sporulation.

Of genes expressed more at hour 24 than hour 0, the one most expressed at hour 24 was *EAH_00011300*. Its expression increased markedly beginning at hour 8 and steadily increased to a peak at hour 24, undergoing a ~15-fold (log2 = 3.9) increase in expression as compared to hour 0 (Table 3). Several of the top 10 genes at T24 (by TPM) also had elevated TPM (>~5,700) but mid-range differential expression vs. T0. However, *EAH_00032920* (encoding a hypothetical protein) was an exception in that while its expression also increased noticeably at T8 (and peaked at T16 and T24), it underwent a >1,000-fold (log2 = 10) increase in expression at T24 as compared to T0 (Table 3). Two other genes in the top 10 TPM at T24 experienced a steady increase in TPM before peaking at T24: *EAH_00007450* (encodes a glyceraldehyde

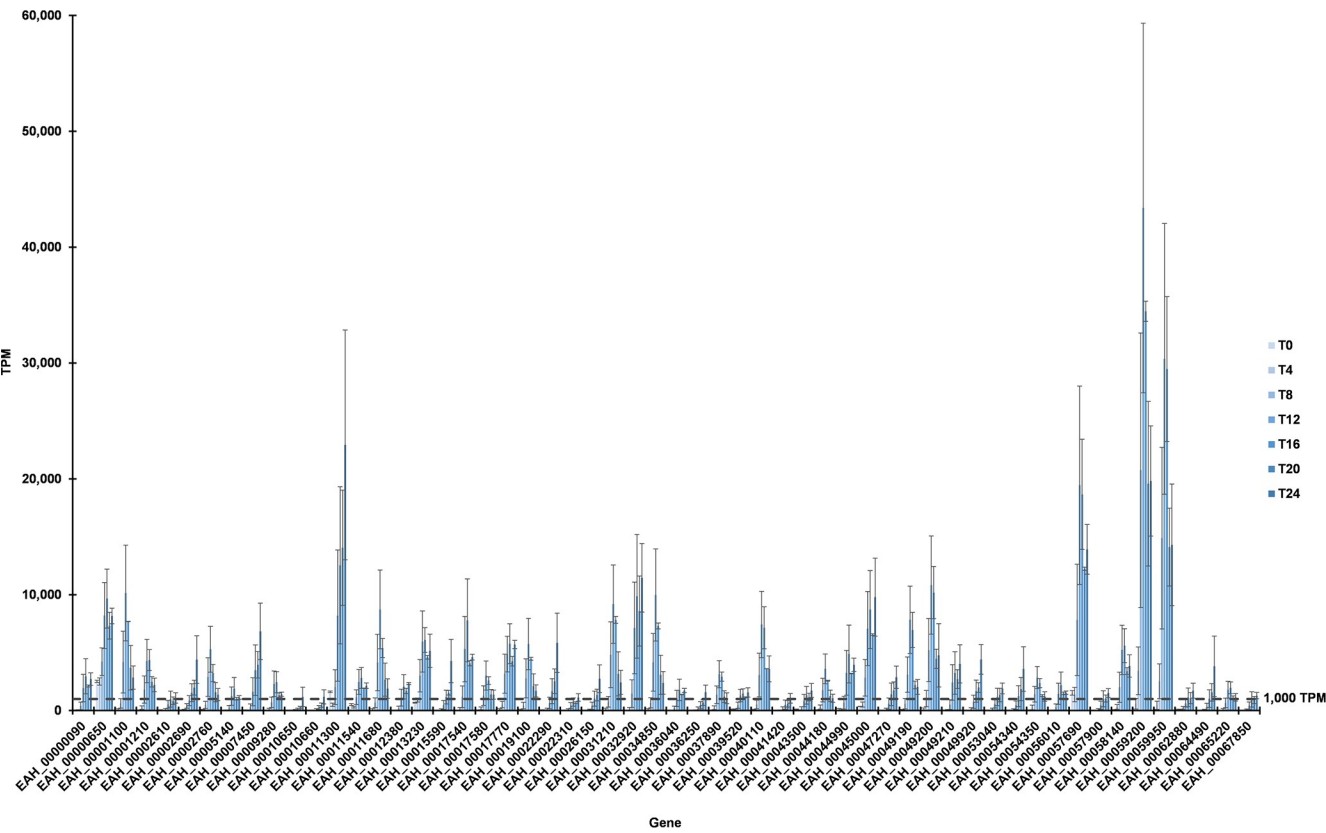

**Fig 8. Significantly up-regulated genes in mature oocysts.** Fifty-six genes contributed at least 1,000 transcripts per million at hour 24 and were transcribed more at hour 24 than at hour 0 (using >1.5 log2 fold change and adjusted *p* <0.05 as threshold criteria).

3-phosphate dehydrogenase) and *EAH_00022290* (encodes a hypothetical protein). These genes were expressed 189-fold (log2 = 7.6) and 127-fold (log2 = 7.0) higher, respectively, than at T0.

Other genes were most-highly expressed at T24 but never reached or greatly exceeded an overall mean of 1,000 TPM (Fig 8, Table 3). Some of these genes (including many encoding hypothetical proteins) contributed at least twice as many TPM at T24 than at any other time. Compared to T0, these genes (*EAH_00002690*, *EAH_00010650*, *EAH_00010660*, *EAH_00015590*, *EAH_00026150*, *EAH_00036250*, *EAH_00049920*, and *EAH_00064490*) were differentially expressed ~84–3,000-fold (log2 = 6.4–11.6). *EAH_00002690* was the only gene meeting these criteria that encoded (by original *E. acervulina* annotation) a non-hypothetical protein (serine protease inhibitor [serpin]).

**Functions of genes expressed most in mature oocysts.** Of the 56 genes identified, 35 (62.5%) were provisionally annotated as hypothetical proteins (S11 Table), exceeding the proportion lacking annotation in constitutively expressed (49%) or initially expressed (50%) genes. Blast2GO found a more informative hit for 17 of the 35 genes lacking any annotation (bolded in S11 Table). The other 18 genes (italicized in S11 Table) either matched to a hypothetical or uncharacterized protein in another species or did not have any other hit (*EAH_00009280*).

Overall, ~66% (37/66) of sequences matched to a protein in *C. cayetanensis*. Of these hits, 18 matched a homologous annotated protein in *Eimeria* or in a related species, 8 matched a different annotated protein, and 11 matched uncharacterized proteins or hypothetical

proteins. With one exception (*EAH_00000090*), uncharacterized/hypothetical protein hits in *C. cayetanensis* matched to original hypothetical protein annotations in *E. acervulina*. With a few exceptions, the "uncharacterized protein" hits matched to original hypothetical protein annotations in *E. acervulina*. For the most highly expressed genes (see above), only a few had homologs that were identified in *C. cayetanensis*, and these were not always informative hits.

In total, 215 GO terms were annotated (S11 and S12 Tables). These were categorized into subontologies of BP (87), MF (73), and CC (55). Forty of the genes (71%) were annotated with at least one term. The genes without GO terms mostly encoded hypothetical proteins. Blast2GO analysis did assign some GO terms to hypothetical proteins, however. Many of the genes strongly expressed at T24 lacked either an informative BLAST hit or GO annotation (other than integral component of membrane, S10 Table). *EAH_00032920* (TPM = 11,461), especially expressed in mature oocysts, had only one homolog as identified by a BLASTP search (a hypothetical protein in *E. maxima*) and zero GO annotations. A few exceptions included *EAH_00002690*, *EAH_00007450*, *EAH_00017540* (a hypothetical protein matching a TRAP-like protein in *Plasmodium chabaudi adami*), *EAH_00049210* (a hypothetical protein matching tubulin-promoting proteins from other species), *EAH_00049920* (a hypothetical protein matching dense granule protein [GRA]12 in *T. gondii*), and *EAH_00064490* (matching a SAG family member protein in *E. mitis*). GO terms associated with these genes included proteolysis, peptidase, and extracellular space (*EAH_00002690*), a number relating to glycolysis and a variety of other biosynthetic processes (*EAH_00007450*), regulation of protein polymerization and microtubule (*EAH_00049210*), and integral component of membrane or cellular anatomical entity (*EAH_00017540*, *EAH_00049920*, *EAH_00064490*). With some exception, most genes encoding hypothetical proteins (S11 Table) were associated with a single GO term grouped in CC sub-ontology (integral membrane component).

Many of the GO terms were assigned to just seven of the 56 genes (*EAH_00000650*, *EAH_00002610*, *EAH_00007450*, *EAH_00047270*, *EAH_00054340*, *EAH_00057900*, *EAH_00067850*). Except for *EAH_00000650* and *EAH_00007450*, these genes had overall moderate or low expression. Within this group of genes (and others), we identified some common functions.

- Some encoded enzymes involved in glycolysis and fermentation (*EAH_00007450*, *EAH_00054340*, *EAH_00057900*, and *EAH_00067850*); these were associated with GO annotations such as carbohydrate and respiration metabolic processes (e.g. glycolysis, gluconeogenesis, pyruvate synthesis, glyceraldehyde-3-phosphate dehydrogenase (NAD$^+$) (phosphorylating) activity, NAD$^+$/NADP$^+$ binding, L-LDH activity, L-malate dehydrogenase activity, fructose-bisphosphate aldolase activity, ATP biosynthesis, ATPase binding, cytosol, apical cytoplasm, actin, nucleotide processes, and others.

- Some mediate stress response and cellular detoxification (*EAH_00002610*, *EAH_00047270*, *EAH_00041420*, *EAH_00044990*, and *EAH_00053040*). These encode heat shock proteins, peroxisomal catalase, peroxiredoxin, superoxide dismutase. GO terms for these genes included responses to stress/chemical, copper ion transport, mitochondrion, cytosol, DNA protection, nucleus, metal ion binding, ATP and ATPase binding, cellular detoxification, response to heat, protein binding and folding, cell redox homeostasis, and thioredoxin peroxidase, and removal of superoxide radicals.

- Additionally, a group of genes (not necessarily highly differentially expressed vs. T0) were involved in actin or myosin processes (*EAH_00000650*, *EAH_00019100*, *EAH_00045000*, *EAH_00057690*) and encoded SAGs (*EAH_00011680*, *EAH_00059200*, *EAH_00059950*).

The former included an assortment of cytoskeleton and organelle GO terms and calcium binding while the latter were associated with integral membrane component.

We found that 9 of the 56 genes (shaded in S11 Table) were classified into KEGG pathways (18 total, Fig 4C). Six of these genes comprised the largest number of GO terms, indicating these genes are involved in well-recognized pathways and functions. Metabolic pathways were enriched for glycolysis/gluconeogenesis (11.4%), carbon fixation in photosynthetic organisms (11.4%), purine metabolism (11.4%), thiamine metabolism (11.4%), methane metabolism (8.6%), glyoxylate and dicarboxylate metabolism (5.7%), pentose phosphate pathway (5.7%), and fructose and mannose metabolism (5.7%). Other pathways encompassing amino acid, fatty acid, and citric acid cycle were also present, but at lower numbers.

## RT-qPCR validated RNA-Seq results

To validate RNA-Seq, we selected six genes for further analysis by RT-qPCR. These genes included some of the most highly expressed genes (two each from the gene lists identified above: constitutively expressed [*EAH_00004110*, *EAH_00033530*], T0 [*EAH_00006770*, *EAH_00050370*], and T24 [*EAH_00007450*, *EAH_00032920*]). Importantly, these genes encompassed a range of TPM and fold changes through time. This enabled us to verify differential expression estimates from RNA-Seq for a wide range of conditions.

For these six genes, we compared expression as estimated from RNA-Seq and RT-qPCR. The log2 FC at each time point estimated by RNA-Seq (using mean of biological replicates for each time point compared to T0) was compared to that found by RT-qPCR (mean of three trials, normalized to T0). As shown in Fig 9A, expression levels determined by RT-qPCR were consistent with those obtained by RNA-Seq. From the RNA-Seq results (Fig 9B, diagonal stripes), *EAH_00004110* fluctuated around +/-1 log2 FC over time relative to T0. *EAH_00033530* (ranged ~0.39–1.8 log2 FC), *EAH_00007450* (ranged 0.32–7.6 log2 FC), *EAH_00032920* (ranged 3.3–10.0 log2 FC), were up-regulated at each time point. Genes *EAH_00006770* (ranged -2.9 - -3.6 log2 FC) and *EAH_00050370* (ranged -1.6 - -2.3 log2 FC) were downregulated. These results show that qPCR validated RNA-Seq as a method to investigate dynamic gene expression as oocysts develop. Furthermore, the methods correlated almost perfectly (Fig 9B, $r$ = 0.985) when applied to the whole dataset.

## Discussion

*Eimeria* are ubiquitous parasites of poultry, causing devastating disease. Control efforts largely depend on vaccines that must be administered as fresh, viable oocysts. Storing vaccines for extended periods can, therefore, undermine immunization efficacy [23]. Various approaches exist to determine apicomplexan oocyst viability *in vitro* [24–40] and *in vivo* [24, 27, 28, 30–32, 35–39, 41–47]. Understanding gene expression in maturing and senescing oocysts might improve upon these available means to evaluate oocyst viability. Molecular methods can also aid assessment of oocyst viability [30, 43–46, 48, 49], and understanding changing gene expression represents a welcome addition to the toolkit. However, all methods have advantages and limitations, which have recently been reviewed in protozoa including the apicomplexans *Cryptosporidium parvum* and *T. gondii* [50].

Better understanding of oocyst development and underlying mechanisms of viability would not only improve vaccine efficacy but could also foster progress in managing the public health risk posed by the agent of human cyclosporiasis. *Cyclospora cayetanensis* is a member of a large, ubiquitous family of coccidian parasites in the phylum Apicomplexa. Phylogenetically, *Cyclospora* may constitute a specialized, morphologically distinct type of *Eimeria* [51, 52] and

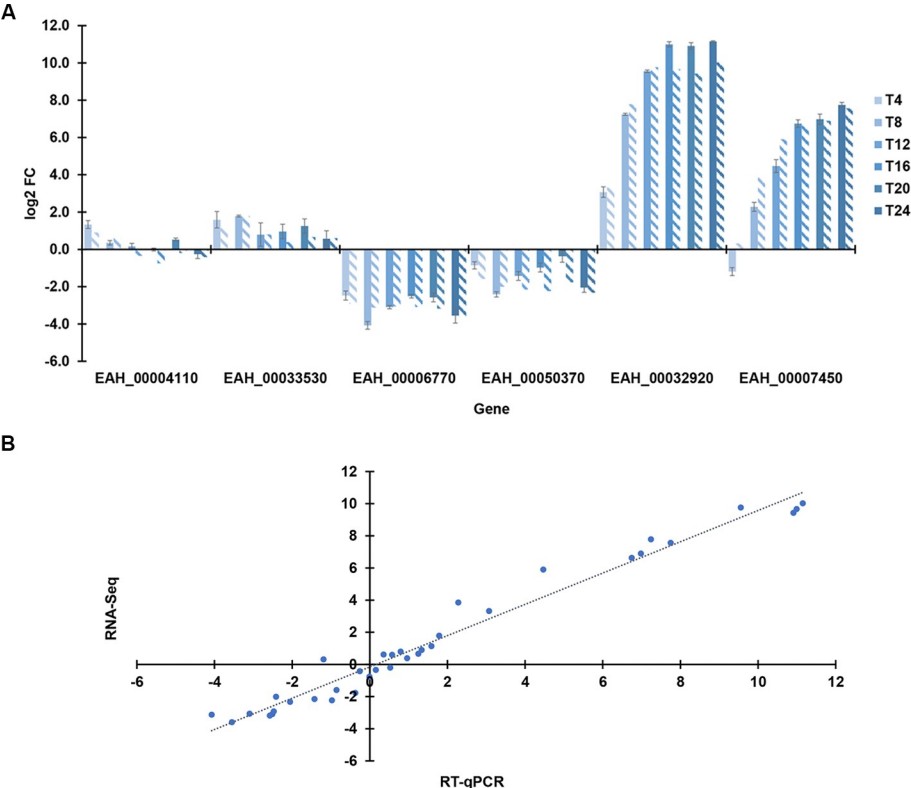

**Fig 9. Highly correlated estimates of transcription from RNA-Seq and RT-qPCR assays.** (A) Temporal changes in expression of genes selected for validation by qPCR. Log2 fold change (FC) of each gene estimated by each method side by side. Bars with diagonal stripes indicate RNA-Seq data. Expression for each time point is depicted relative to initial transcription level (T0). Note that these six genes include ones with stable, decreasing, and increasing expression through time. Error bars depict variance among RT-qPCR estimates among biological replicates. (B) Correlation of expression as estimated from RT-qPCR and RNA-Seq for six genes at each of six time points. Gray dots indicate regression line.

more recent studies have further shown the close relationship between *C. cayetanensis* and avian-infecting *Eimeria* spp. [53–55]. Progress in understanding and managing *C. cayetanensis* has been hindered by the fact that it infects only humans and lacks a suitable animal model [56]. Therefore, researchers have turned to surrogate organisms, including *E. acervulina* and *T. gondii* [35, 36, 42, 57, 58]. *Eimeria acervulina* is intriguing as a surrogate because it poses no occupational safety risk to researchers, it has a low infectious dose, and is relatively easy to passage *in vivo* and produce many oocysts. Although sensitive methods exist to detect *C. cayetanensis* oocyst contamination in food and water matrices [59–62], a means to determine oocyst viability (and thus, infection risk oocysts pose to public health) remains elusive. Our transcriptome analysis is ideal for addressing this issue. In addition to finding putative markers of viable oocysts and differentially expressed genes demarcating immature and mature oocysts, we defined temporal changes during sporulation. Furthermore, functional analysis allowed us to identify homologs of these markers in other apicomplexans, including *C. cayetanensis*.

## Constitutively expressed genes as putative biomarkers of viability

We identified over 50 genes expressed constitutively throughout oocyst sporulation, some of which contributed to the highest proportion of transcripts (as estimated by TPM). These genes

include those with general housekeeping functions. The consistent expression of these genes suggests their utility as markers of viability.

**ATPases with possible mitochondrial function.** The two proteins with homology to Cation-transporting ATPases (encoded by EAH_00004100 and *EAH_00004110*) matched most to an *E. acervulina* sequence (named *Eimeria*-conserved protein [63]) and an *E. maxima* Cation transporting ATPase (EMWEY_00006260), respectively. We found matches for each in *C. cayetanensis*, but function was not obvious to interpret as the matched genes encoded "low quality proteins". The *E. tenella* homolog of *EAH_00004100* was found to be developmentally expressed, with highest gene expression in sporozoites; expression was also higher in mature oocysts than immature oocysts [63]. Furthermore, expression levels of this ATPase in mature oocysts were downregulated in precocious lines of *E. maxima* [64] and *E. tenella* [63], indicating it could be important for parasite development. Our results differed from these studies, as we observed highest expression for both genes during early oocyst development (T4 and T0). Both genes matched another *E. tenella* gene (*ETH_00003230*, encodes a hypothetical protein) listed as an ortholog/paralog in ToxoDB and it was shown to be highest expressed in immature oocysts [12]. GO terms identified for *EAH_00004100* and *EAH_00004110* included integral component of membrane (both) and mitochondrion (*EAH_00004100*). These P-type ATPases use the energy of ATP hydrolysis to transport cations and lipids across membranes. In apicomplexans, P-type ATPases may be responsible for transporting Calcium, which is critical to a number of cellular functions, including parasite growth and differentiation [65]. Mitochondria are known calcium stores. We cannot yet state conclusively a role for these ATPases in sporulation, but these data may indicate an important role for ATPases and mitochondrial function in viable oocysts.

**Oocyst Wall Proteins (OWPs) and palmitoyltransferases.** *EAH_00033530* was expressed most in T8 oocysts, maintaining strong expression throughout sporulation. An *E. tenella* gene (*ETH_00025735*) encoding a homologous OWP was reported to be up-regulated in immature oocysts [12]. OWPs in *Eimeria* spp. were also reported to be expressed most in gametocytes [66, 67] but studies in *T. gondii* [68] and *E. tenella* [69] found elevated expression of OWPs (or a protein matching to OWPs in other species) in mature oocysts. It is unclear why *EAH_00033530* fluctuated in expression throughout the sporulation course, with highest expression earlier in sporulation. It is possible that sporozoite and sporocyst synthesis increases early and then stabilizes once sporocysts are fully formed in mature oocysts. Another interesting up-regulated gene was *EAH_00034270* (encodes a zinc finger DHHC domain-containing protein), which had very high expression early in sporulation (T4). It matched to a similar protein in *E. maxima* and *T. gondii* and also matched to palmitoyltransferases in other apicomplexans, including *C. cayetanensis* and *Plasmodium* spp. DHHC domains are found in palmitoyltransferases, which localize to the inner membrane complex (IMC) in apicomplexan parasites such as *Plasmodium falciparum* and *T. gondii*, and palmitoylated proteins play key roles in the organization and function of the IMC [70, 71]. Perhaps the early and continued upregulation in oocysts is indicative of macromolecule modification and cellular organization important for growth and development.

## Developmentally-distinct gene expression and concerted expression changes demarcating sporulation

Comparing constitutively expressed genes and those differentially expressed between immature and mature oocysts revealed that distinct expression profiles typify each phase of development, and such profiles have been reported previously among *Eimeria* spp. life cycle stages. In *E. tenella*, sporozoites and sporulated oocysts share many common transcripts, whereas immature oocysts have many transcripts not shared by either sporozoites or merozoites [72]. Similar

developmental patterns occurred in immature, maturing, and mature oocyst stages of *E. acervulina*, *E. maxima*, and *E. tenella* [73]: sporulating oocysts more resembled unsporulated oocysts, whereas mature oocysts more closely resembled sporozoites. These results are in line with those described by Lal et al. [74], who reported on proteins expressed in *E. tenella* stages. Transcriptomes of *C. parvum* oocysts and sporozoites are also related [75].

We affirmed studies, of other species of *Eimeria*, determining that genes defining sporulation predominate about 12 hours into sporulation [76–78]. We observed a shift in expression patterns between hours 8 and 16. Some of the most variable genes could clearly be attributed to stages of development. Some of the highest-expressed genes in mature oocysts monotonically increased toward T24. We also identified genes that increase in expression early in development. For example, certain constitutively-expressed genes peak early and undergo a subsequent expression peak around T20. For some genes expressed abundantly in immature oocysts, expression subsequently ebbed and then increased, never regaining their prior peak levels. These results may indicate that sporulation is asynchronous, an observation also made in other *Eimeria* spp. [73], or that distinct expression profiles may typify each phase of development.

### Transcripts characteristic of immature oocysts

Our analysis enabled the identification of a few genes most characteristic of immature oocysts. The two highest-expressed genes (*EAH_00006770*, *EAH_00050370*), which were annotated as encoding hypothetical proteins in *E. acervulina* had homology only to genes encoding hypothetical proteins in other species. An *E. tenella* gene (*ETH_00023355)* encoding a similar hypothetical protein was also reported to be highly expressed in immature oocysts [12]. Their conserved nature of these genes beckons further analysis to characterize function, as they may define a signature for young oocysts.

**Early expression of a subtilase family serine protease.**   We identified a serine protease (encoded by *EAH_00048170*) with similarity to subtilisins in other apicomplexans. Subtilisins are related to subtilases, and the latter are involved in processing secreted proteins in apicomplexans, having roles in secretory organelle maturation, assembly and trafficking of organellar content proteins, invasion, and egress processes [79, 80]. Subtilisins are expressed in different stages of the *E. tenella* life cycle (including throughout oocyst sporulation [81]) and some can be expressed higher in immature oocysts [67, 81, 82]. These studies showed subtilisins in *Eimeria* spp. likely contribute to oocyst wall formation, microneme processing, and cell invasion in different life stages. We found *EAH_00048170* was most expressed at T0 (with a secondary peak at T20 and with decreased expression in mature oocysts). Perhaps this protein is important for early oocyst developmental processes packaging sporozoites but it becomes down-regulated as oocysts progress to a fully mature and dormant state. A gene encoding a homologous subtilisin-like protein in *E. tenella* (*ETH_00005950*) was also up-regulated in immature oocysts [12], indicating this type of protease is likely important for immature oocysts.

**Metabolic enzymes especially characteristic of immature oocysts.**   Two other genes encoded or matched to important metabolic enzymes.

- *EAH_00031010* encodes NADP-specific glutamate dehydrogenase, and KEGG analysis found it functions in amino acid and nitrogen metabolism. A gene encoding a homologous protein in *E. tenella* (*ETH_00033375*) was up-regulated in immature oocysts [12]. Glutamate dehydrogenases were found to localize to the apicoplast or cytosol in *P. falciparum* [83]. Therefore, it is possible this enzyme acts in a similar way in *Eimeria* oocysts, but we cannot yet state a definitive role for this gene.

- *EAH_00034940* encodes a hypothetical protein and a gene encoding a homologous hypothetical protein in *E. tenella* (*ETH_00007785*) was also up-regulated in immature oocsts [12]. Each protein resembled LDH in *E. necatrix* and peroxisomal alkyldihydroxyacetone-phosphate synthase in *C. cayetanensis*. The GO terms and KEGG pathway identified for *EAH_00034940* seemed to apply to the alkyldihydroxyacetonephosphate synthase hit, which is a peroxisomal enzyme involved in ether lipid biosynthesis. This enzyme was found to be upregulated in *E. maxima* gametocytes, characterized as an oxidoreductase linked to oocyst wall biosynthesis [67]. As described above, we observed high expression of an OWP early during sporulation. As ether lipids are constituents of cell membranes, perhaps this enzyme is necessary for early oocyst structure formation.

**Other genes characteristic of immature oocysts may mediate protein synthesis, organization, compartmentalization, cleaving, and packaging.** Three more genes in immature oocysts, while having more muted expression, were still expressed >3 times higher than T24 oocysts (Table 3).

- *EAH_00000360* encodes a hypothetical protein and had informative homology only to an *E. tenella* secreted protein (with signal peptide, 12 KAZAL repeats, and a mucin-like stretch of threonines, *ETH_00006770*). This gene was up-regulated in unsporulated *E. tenella* oocysts [12]. *C. parvum* oocysts contained a secreted glycoprotein with 12 KAZAL repeats that was rich in serine and threonine [84]. While this protein was not directly implicated, other mucins associate with the sporozoite surface and tether them to the inner surface of oocyst walls. A mucin-associated surface protein was also found exclusively in early oocysts in *E. tenella* [74]. It is possible that the protein we identified may be involved in very early sporocyst and sporozoite synthesis and anchoring.

- *EAH_00020350* encodes a haloacid dehalogenase-like hydrolase domain-containing protein. These proteins are a large superfamily of phosphatases present in eukaryotes that possess a plastid-like organelle (*e.g.* plants and apicomplexans). In *P. falciparum*, they are known to regulate metabolic processes and factor in homeostasis [85]. A gene encoding a homologous protein in *E. tenella* (ETH_00027645) was reported to be expressed more in mature vs. immature oocysts [12] and to a great extent in gametocytes [67]. A gene encoding a protein with a haloacid dehalogenase-like hydrolase domain was expressed comparatively higher in immature *T. gondii* oocysts [68]. This protein also matched a chloroplastic endoribonuclease YBEY in *C. cayetanensis*. This is likely because plant endoribonuclease YBEY proteins contain haloacid dehalogenase domains [86] and apicoplasts are non-photosynthetic chloroplast-like organelles. The significance of this gene in oocysts remains to be determined.

- *EAH_00002160* encodes a microtubule-binding protein and similar proteins were identified in other *Eimeria* spp. as well as autophagy-related proteins in *C. cayetanensis* and *T. gondii*. Autophagy facilitates cellular homeostasis, promotes cell survival by eliminating/recycling damaged organelles, and is important for invasion in apicomplexan parasites [87]. Microtubules factor importantly in autophagy, mainly in the formation and motility of autophagosomes [88]. The related autophagy protein in *T. gondii* (ATG8) plays essential roles in autophagosome biogenesis and the inheritance of apicoplasts during cell division [89]. ATG8 in *E. tenella* sporozoites activates upon starvation [90]. Deletion of another autophagy protein in *T. gondii* (ATG3) inhibits conjugation of ATG8 to the autophagosome, thereby disrupting mitochondrial homeostasis and normal growth of tachyzoites [91]. A gene encoding a homologous microtubial-binding protein in *E. tenella* (*ETH_00016760*) was expressed in multiple life cycle stages, but highest in immature oocysts [12]. Therefore, the

upregulation of microtubule proteins facilitating autophagy in immature oocysts may indicate autophagy plays important roles in oocyst development.

These data identify genes supporting metabolic processes that evidently contribute to young oocyst growth and development. Immature oocysts appear to activate distinct processes from those in mature oocysts; early developmental processes may prepare oocysts for subsequent division and differentiation into sporocysts and sporozoites. These processes likely require protein synthesis, organization, compartmentalization, cleaving, and packaging. Similar protein and gene expression profiles occur in other species of *Eimeria* spp. [74, 92–94] and in *T. gondii* [68], indicating broadly conserved developmental processes in Apicomplexa, writ large.

### Transcripts characteristic of mature oocysts

**Genes most expressed in mature oocysts appear least phylogenetically conserved and may demarcate functional specialization.** We identified 56 genes most expressed in mature oocysts. The most highly-expressed gene in mature oocysts (*EAH_00011300)* encodes a hypothetical protein resembling a late embryogenesis abundant (LEA) protein in *C. cayetanensis*. LEA proteins are associated with cellular protection from water loss and cold stress in plants and animals [95]. A gene encoding a similar protein in *E. tenella* (*ETH_00030440*) was also expressed highest in mature oocysts [12]. The protein also had similarity to *T. gondii* LEA orthologs and paralogs in ToxoDB. Four LEA-encoding genes increased expression throughout sporulation in *T. gondii* oocysts [68].

The function of another major transcript in mature oocysts (*EAH_00032920*), which undergoes more >1,000-fold increase in expression (Table 3), remains unclear. Functional analysis, even with the most relaxed BLAST settings, identified only a single match to a hypothetical protein in *E. maxima*. Because it appears to be an important transcript in mature oocysts, we intend to further characterize it and determine if it serves as a marker of viability

In comparison to constitutively-expressed genes and those characterizing immature oocysts, genes characterizing mature oocysts were least annotated. Many of these lacked informative matches in other species, including *C. cayetanensis*. Acknowledging the rudimentary state of the current genome annotation, it is reasonable to surmise that mature oocysts may tend to increase expression of genes encoding proteins unique to apicomplexans. Similarly, a study of *C. parvum* concluded that oocysts (and sporozoites) preferentially express genes encoding specialized functions which have few orthologs outside of related protozoa [75].

**Metabolic enzymes characteristic of mature oocysts.** We identified four genes (*EAH_00007450*, *EAH_00054340*, *EAH_00057900*, and *EAH_00067850*) that steadily increased in expression in maturing oocysts and encode enzymes involved in major energy and metabolic pathways. These enzymes (glyceraldehyde-3-phosphate dehydrogenase, LDH, and two Fructose-bisphosphate aldolases, respectively) are key components of processes that likely provide energy stores to sporozoites, including glycolysis and gluconeogenesis glycolysis/gluconeogenesis, methane metabolism, the pentose phosphate pathway, and fructose and mannose metabolism. Amylopectin (a branched polysaccharide composed of glucose) accumulates in coccidian oocysts and serves as an energy source for excystation and subsequent penetration of cells [96, 97]. In fact, in *E. acervulina*, amylopectin is utilized by dormant sporozoites in mature oocysts and as levels decreased, oocysts were less viable and less infective *in vivo* [96]. Upregulated gluconeogenesis enzymes may be necessary to maintain glucose and subsequent amylopectin stores.

Expression of *LDH* (encoded by *EAH_00054340*) increased more than 600-fold in mature oocysts when compared to hour 0 (Table 3). *LDH* expression in *E. tenella* was found to be

restricted to schizonts; protein levels did not differ among various stages (including oocysts assessed at multiple time points during sporulation) [98]. Other *E. tenella* studies showed LDH protein levels were similar in oocyst stages but transcription levels of *LDH* were lower in mature vs. immature oocysts (although highest expressed in merozoites) [12, 99]. High *LDH* expression has been reported in mature oocysts in another coccidian, *C. parvum* [75, 100]. An *E. maxima* study showed that LDH and fructose bisphosphate aldolase in sporozoites bound chicken jejunal epithelial cells [101] and it is known that immunization with *E. acervulina* LDH can induce protective immunity [102]. Fructose bisphosphate aldolase was found to be important for energy production and invasion in *T. gondii* [103] and *P. falciparum* [104]. We found that two genes encoding aldolase enzymes were upregulated in mature oocysts and expression began to increase around 12 hours during sporulation. Of the two genes encoding aldolase homologs in *E. tenella* (*ETH_00008600*, *ETH_00015505*), *ETH_00008600* peaked in expression in sporulated oocysts [12]. These data indicate the enzymes are important in maturing oocysts.

Clearly, certain energy-synthesizing pathways (aerobic and anaerobic) are important to sporulation. Many enzymes involved in energy metabolism pathways, such as glycolysis and fermentation, were highly expressed in *C. parvum* sporulated oocysts [100]. In *E. tenella*, RNA-Seq characterized gene expression in oocysts (0–96 hrs of development) and sporozoites, documenting increased expression of some metabolic enzymes (integral to glycolysis, the citric acid cycle, and pentose phosphate pathways) in sporulated oocysts (48 h) [92]. A proteomic study, also conducted in *E. tenella*, found several proteins involved in metabolic pathways (glycolysis, gluconeogenesis, amino acid metabolism and lipid metabolism) in merozoites, sporozoites, and oocysts [74]; no appreciable difference between immature and mature oocysts was noted. Proteins were most numerous in oocyst stages, particularly enriched for those involved in glycolysis and gluconeogenesis. Interestingly, enzymes involved in the mannitol cycle were found in all stages, except merozoites. Mannitol is known to concentrate at high levels in immature oocysts and it is thought to be a main energy source necessary for sporulation [105]. Here, we found the only *E. acervulina* gene annotated to encode a mannitol cycle enzyme (*EAH_00033080*, Mannitol-1-phosphate dehydrogenase) is significantly up-regulated in mature oocysts but never reached high levels of expression (<100 TPM throughout sporulation, S2 Table).

**Transcription factors.** In our search for markers of viability, we hypothesized that TFs might orchestrate sporulation. A class of TFs unique to Apicomplexa (ApiAP2) was first reported by Balaji et al [22]. They carry a domain with similarity to an Apetala2/ethylene response factor (AP2/ERF) integrase DNA binding domain, present in plants. ApiAP2 TFs play critical roles in for parasite differentiation, development, and virulence [18, 106–112]. Most of these studies were in *Plasmodium* and *Toxoplasma* spp.; much less is known in other apicomplexan species. In *Eimeria* spp., 44–54 genes containing ApiAP2 domains have been identified. These include 21 *Eimeria*-specific ApiAP2 groups, 22 groups shared with other coccidia, and five pan-apicomplexan clusters [12]. Genes containing ApiAP2 domains were previously found up-regulated in *E. necatrix* gametocytes and merozoite stages [113, 114]. This difference in expression profiles for ApiAP2 domain-encoding genes was associated with the developmental fates of second- and third-generation merozoites. Three genes encoding hypothetical proteins (with similarity to AP2 domain TFs) were upregulated in *E. tenella* gametocytes [67].

Here, we identified low expressed genes annotated as generic TFs but some were significantly expressed compared to T0 oocysts. We also focused on *T. gondii* ApiAP2 homologs in *E. acervulina* and found 49 unique matches (annotated mostly as hypothetical proteins but having GO annotations for regulation of transcription). While most were also expressed at a

low level, several were significantly differentially expressed between immature and mature oocysts. Recently, CRISPR-Cas9 was used to disrupt the repertoire of ApiAP2 genes in *E. tenella* [110]. Systematic targeting of the 33 ApiAP2 TFs found that 23 are essential for development and survival in the host. A cursory analysis of transcription data [12] for these TFs in ToxoDB found many were differentially expressed between immature and mature oocysts. Clearly, ApiAP2 TFs play important roles in *Eimeria* spp. and more studies are required to assess the roles they play in oocyst development. Their low but stage-specific expression in sporulating oocysts make them intriguing for markers of development and potentially markers of viability.

**Surface proteins and secretory organelle proteins upregulated in mature oocysts.** The high expression of secretory and surface proteins, and serpins in sporulating oocysts has not only been captured in our data but also reported by others. Micronemes and GRAs (dense granule proteins) are part of the apical complex of organelles; proteins secreted from these organelles enable host invasion by promoting parasite motility, host cell adhesion, and host cell remodeling [80, 115]. Some surface antigens (termed Surface Antigen Related Sequences [SRSs]), GRAs, and micronemes were up-regulated in *T. gondii* sporulating oocysts, indicating induction as sporozoites begin to form [68].

SAGs are abundant glycosylphosphatidylinositol-anchored molecules on the surface of invasive stages of apicomplexan parasites. SAGs elicit strong immune responses and promote ligand binding, -zoite maturation, host cell invasion, immune evasion, and pathogenicity [116, 117]. SAGs are the principal surface antigen gene family in *E. tenella* and are differentially expressed in a variety of *E. tenella* stages, including immature and mature oocysts [12, 117]. The major SAG genes in oocysts are of the *sagA* type, common in all *Eimeria* spp. These genes may promote attachment and facilitate invasion of host cells [12]. We observed three genes encoding SAGs (*EAH_00011680*, *EAH_00059200*, *EAH_00059950*) and another gene (*EAH_00064490*) encoding a protein resembling a SAG protein in *E. mitis*. Two of these were among the top three highest-expressed genes (by TPM) in mature oocysts and *EAH_00064490* was expressed over 1900-fold compared to immature oocysts. The SAGs we identified matched only to SAGs in other *Eimeria* spp. and *C. cayetanensis*, suggesting these proteins are characteristic of these eimeriid parasites. SAGs in *Eimeria* are different than the predominant surface antigens families in the apicomplexans *Toxoplasma* and *Plasmodium* [12]. Also, a recent study showed that a fold within a representative SAG[B] subfamily protein in *E. tenella* are distinct from surface antigens of *Toxoplasma* and *Plasmodium*, and that this fold defines other *Eimeria* SAG sub-families [118]. Furthermore, this fold is conserved in *Eimeria* and *C. cayetanensis* and unrelated to SAGs or SRS proteins in other cyst-forming coccidia.

Miska et al. [72] identified an abundant transcript from *E. tenella* sporulated oocysts encoding the microneme protein Etmic-1. In both virulent and precocious *E. maxima* strains, RNA-Seq found genes encoding microneme, rhoptry (another apical complex organelle), and IMC proteins were up-regulated in mature oocysts [94]. In addition, one GRA and over 20 SAGs were upregulated in mature oocysts. Abundant transcripts for microneme proteins (also a protein disulfide isomerase-like protein and SAGs) were also reported in *E. tenella* sporulated oocysts [69]. Microneme proteins (MIC1-5, MIC7) were also detected in mature *E. tenella* oocysts but absent from early oocysts [74]. Additionally, expression of EtMIC1-5 is highly coordinated during sporulation; these proteins were detected only after sporozoites began to mature. Interestingly, mRNA levels for all of these genes were detected 10–12 hr earlier in sporulation [119]. These data suggest that microneme protein expression is regulated at both the transcriptional and translational levels and this coincides with sporozoite development.

We found similar timing of microneme expression here:

- *EAH_00000090* and *EAH_00017580* were minimally expressed at T0, and peaked at T12-T16. The genes were expressed over 40-fold in mature vs. immature oocysts. A single homolog to *EAH_0000090* was identified in *E. tenella* (Etmic-2, encoded by *ETH_00006930*) and it was expressed in mature oocysts, sporozoites, and merozoites with no expression in immature oocysts [12]. Microneme proteins in *Eimeria* spp. sporozoites interact with chicken jejunal cells, indicating an important binding role to cells [101, 120]. Therefore, the upregulation of micronemes in mature oocysts may indicate that fully-formed packaged sporozoites express these proteins, necessary for invasion.

We also observed that a gene (*EAH_00049920*) resembling a GRA in *T. gondii* (GRA12) was expressed over 2000-fold in mature vs. immature oocysts. Two genes encoding homologs (*ETH_00023950*, *ETH_00024035*, hypothetical proteins) in *E. tenella* were expressed highly in mature oocysts and sporozoites (highest) and not expressed in immature oocysts [12]. GRAs are secreted during parasite entry into a host cell and likely serve a role in remodeling the parasitophorous vacuole and maintaining the host-parasite relationship [121]. The functions of GRAs are not completely understood, best characterized in *Sarcocystis* and *Toxoplasma* spp. (~20 GRAs in *Toxoplasma*). With little known about GRAs in *Eimeria* spp., one study attempted expression of GRA7 from *T. gondii* in *E. tenella* sporozoites [122]. Mature oocysts from infected birds expressed the protein within sporocysts. In *T. gondii*, GRA12 helps establish infection in host cells, aids immune system evasion, and modulates pathogenesis. GRA12 mutants decrease virulence and tissue cyst burden *in vivo* and are more easily eliminated by host immunity [123]. Recently, the *E. tenella* GRA9 protein was tracked during parasite development using a CRISPR-Cas9 approach [110]. It was found to be a secreted protein that may assist sporozoite release from sporocysts and it may be important during invasion by sporozoites and merozoites. The high expression of *EAH_00049920* in mature oocysts in *E. acervulina* indicates GRA or GRA-related proteins are important for similar functions.

The cellular localization and function of these apical and surface proteins deserve confirmation, as microneme, SAGs and GRA proteins appear important as markers for mature oocysts, promoting host cell recognition, attachment, and invasion. As with other upregulated genes identified here, we believe expression indicates stocking elements of sporozoites that will be packaged in sporocysts and eventually emerge read to invade cells.

**Serpins.** Serpins are a large family of proteins that inhibit serine proteases and regulate proteolytic cascades [124] essential to many physiological processes. The two serpin-encoding genes we identified (*EAH_00001210*, *EAH_00002690*) were up-regulated at least 80-fold in mature vs. immature oocysts. While one study in *E. tenella* reported a serpin was up-regulated in mature oocysts [72], serpins are not exclusively expressed in mature oocysts of *E. maxima* [64], *E. acervulina* [124], and *E. tenella* [125]. Inhibiting serine proteases can prevent invasion of apicomplexan (including *Eimeria*) parasites into host cells [79, 81] and inhibition of serine/cysteine proteases prevents processing of GAM56, essential for incorporating peptides into the oocyst wall, in gametocytes of *E. tenella* [82]. An *E. tenella* serpin (encoded by *ETH_00011330*) homologous to *EAH_00002690* was up-regulated in mature oocysts and sporozoites (higher in sporozoites) [12]. A serpin was also upregulated in maturing *T. gondii* oocysts [68]. Two GO terms pertaining to metalloendopeptidase inhibition were associated with *EAH_00001210*. Metalloproteases are responsible for growth, replication, and invasion/egress in *P. falciparum* and *T. gondii* [80] and are important for development in *E. tenella* [126]. The peak expression of this gene midway through sporulation may indicate a balance of growth promotion and inhibition is required during parasite development. The range of functions of serpins, including their role in sporulation, remain ill-defined and serpins merit further investigation.

**Glideosomal proteins.** A group of genes encoding actin (and actin regulation) and myosin proteins were also up-regulated in mature oocysts. These included a few of the top highest-expressed genes we identified in mature oocysts (*EAH_00000650* [actin], *EAH_00045000* [actin depolymerizing factor], and *EAH_00057690* [profilin]) as well as *EAH_00019100* (myosin). *EAH_00000650* was also a gene we identified as constitutively expressed and it is associated with a host of GO terms and some metabolic pathways. While it increases expression in mid to late oocysts, it is not greatly differentially expressed vs. T0 oocysts. In comparison, actin-facilitating proteins such as *EAH_00045000* and *EAH_00057690* have greater TPM throughout the time course and are expressed at least 10-fold greater in mature oocysts. *EAH_00019100* was not as highly expressed as the other genes, but it still was differentially expressed 24 times greater in mature vs. immature oocysts.

These proteins are known to make up the "glideosome", responsible for gliding motility and invasion in extracellular stages of apicomplexan parasites [127, 128]. Several we identified were expressed in various stages of *E. tenella* (including multiple oocyst developmental stages) [12, 74, 129]. Myosins were also upregulated in sporulated oocysts in an *E. maxima* RNA-Seq study [94]. Gliding motility and invasion are intricately tied together in a process that involves microneme secretion and actin-based movement of the glideosome. The glideosome has been characterized in *T. gondii* and this model likely applies to other apicomplexans. It is made up of myosin and other proteins and is anchored to the outer membrane of the IMC and connected via actin and aldolase to a microneme protein complex, which, in turn, interacts with host-cell receptors [127, 128, 130]. Interestingly, we identified two aldolase enzymes (described above) up-regulated in mature oocysts. Both enzymes were expressed relatively low (by TPM) but steadily increased to T24. Each gene was differentially expressed >100 fold in mature vs. immature oocysts, however. Therefore, it is possible the expression of these enzymes factor in the bridging role of the glideosome in addition to glycolysis. The higher expression we noted for these proteins in mature oocysts may indicate important cytoskeletal involvement coinciding with sporozoite maturation, packaging, preparation for excystation. But general expression throughout could indicate parasite maintenance and additional roles in oocysts.

**Stress response genes.** Finally, we identified increasing expression of genes encoding proteins related to protein folding, stress (heat, oxidative), antioxidants, and detoxification- heat shock proteins (HSPs, *EAH_00002610*, *EAH_00044990*), peroxisomal catalase (*EAH_00041420*), peroxiredoxin (*EAH_00047270*), and superoxide dismutase (*EAH_00053040*). Catalase, peroxiredoxin, and superoxide dismutase are enzymes that protect cells against radical oxygen species. All these genes were expressed comparatively higher in mature oocysts vs. immature oocysts, ranging from ~26-fold (HSPs) to at least 111-fold expression with the others. Two genes encoded enzymes mapping to KEGG pathways: thiamine and purine metabolism (*EAH_00002610*) and tryptophan metabolism and Glyoxylate and dicarboxylate metabolism (*EAH_00041420*). Tryptophan can act as an antioxidant and remove reactive oxygen species and enhance resistance of the damage caused by free radicals [131]. A glyoxylate shunt is activated in response to oxidative stress in some organisms but it is not thought to be present in apicomplexans such as *Eimeria* spp., *P. falciparum* and *T. gondii* [132]. The up-regulation of these types of genes are likely interlinked and individual roles may be hard to discern.

*Toxoplasma* have enzymes associated with detoxification of reactive oxygen species and antioxidant proteins such as glutaredoxin, a putative glutathione/thioredoxin peroxidase and a superoxide dismutase) were upregulated in *T. gondii* sporulated oocysts [68, 133]. *E. tenella* encodes homologous enzymes and two (matching peroxisomal catalase and superoxide dismutase) were reported to be up-regulated in mature oocysts [12]. However, superoxide dismutase and catalase activity were also shown to be low (or comparatively lower to other

stages) in mature *E. tenella* oocysts [12, 134]. Multiple genes encode these important enzymes, so these may exist to be expressed at various stages when needed in development.

HSPs are ubiquitous and a part of the cellular stress response, facilitating protein folding. Expression of HSPs increases during stress conditions such as heat, nutrient deprivation, or chemical exposure [135]. Indeed, HSPs may be activated under conditions that promote excystation of sporozoites (mechanical oocyst disruption or incubation in trypsin or bile salts [136]). Therefore, the conditions we subjected oocysts to may have caused the upregulation of these genes. *EAH_00002610* is annotated as a generic HSP but has homology to HSP70 in other apicomplexans, including *C. cayetanensis*. A gene encoding a homologous protein in *E. tenella* (ETH_00000210, hypothetical protein) was also upregulated in sporulated oocysts [12]. HSP70 is involved in development, differentiation, infectivity, and virulence in multiple parasites, including Coccidia [137–139]. HSP70 protein content accumulates in *E. tenella* during sporulation (sporoblasts, sporocysts, and sporozoites in late oocysts) but dissipates by the time oocysts are fully developed [136]. It was theorized that HSP70 activity is involved in the development of sporocysts and sporozoites. A subsequent study found inhibiting HSP70 in oocysts interfered with synaptonemal complex formation and inhibited sporulation [140].

*EAH_00044990* encodes a protein with a HSP20/alpha crystallin domain (characteristic of small heat shock proteins [sHSPs]). It had mid-range TPM in mature oocysts and expression peaked at T16, indicating a role for this protein in growth, likely around the time when sporocysts are maturing and sporozoites are under development. Homologs encoded by *E. tenella* (*ETH_00017510*, hypothetical protein; *ETH_00026340*, HSP28; and *ETH_00027135*, HSP20) were upregulated in sporulated oocysts [12]. sHSPs are regulated at different parasite developmental stages and likely play roles in parasite differentiation (including apicomplexans) like HSPs [141, 142]. HSP20 was shown to be important for *Plasmodium* sporozoite adhesion and migration motility [143]. Furthermore, HSP20 may regulate several aspects of actin activity in *Plasmodium* and other species, including binding and polymerization [144]. *EtsHsp20.4* in *E. tenella* was expressed in different parasite stages and it was highest in mature oocysts [139]. An *E. tenella* protein resembling HSP21 in *T. gondii* was up-regulated in sporulated oocysts [69]. Also, HSP20 was expressed more in mature than in immature *T. gondii* oocysts [68].

## Conclusions

Mature oocysts especially express genes that support respiration, carbon fixation, energy utilization, and related processes. Oocyst maturation is defined by these metabolic processes as well as by stress responses. In addition, genes implicated in oocyst sporulation, growth, development, and potentially invasion of the host cell and host-parasite interactions are upregulated in mature oocysts. As sporozoites are packaged and prepared for motility and harsh conditions in the gut, they store energy [94] and prepare proteins required for invasion, proliferation, and division. We identified constitutively expressed genes involved in a variety of basic functions and genes that allowed the discrimination of immature and mature oocysts. We more fully characterized the process of oocyst maturation for the purpose of developing, validating, and applying biomarkers demarcating viable, infectious oocysts.

Conclusively establishing the function for some of the most highly-expressed and differentially-regulated genes awaits progress in coccidian genome annotation. Such progress will enrich the understanding of mechanisms underlying sporulation and viability, in *Eimeria* and in related taxa, such as *C. cayetanensis*. With recent sequencing completed on *Eimeria* spp. [12] and isolates of *C. cayetanensis* [53, 55, 145–152], a growing fund of genomic data support homology studies. Such data will enhance efforts to understand the role genes we identified, elucidating key pathways supporting sporulation, invasion, growth, and development.

Such data portend many practical applications. For example, these results suggest transcripts that may serve as informative targets to rapidly detect viable (and infectious) oocysts, enhancing *Eimeria* vaccination efforts and enabling researchers, regulators, and growers to improve produce safety through improved control of *C. cayetanensis*.

Extending such studies to sporocyst excystation and invasion will further establish biomarkers for viability, as not all mature oocyst cohorts, enriched with sporulated oocysts, are infectious. Pairing gene expression with functional assays of infection in poultry could substantially contribute to risk assessment, novel treatments, and mitigation in *C. cayetanensis*, for which no animal or cell culture propagation system yet exists. Such applications await determination of the long-term fate of these transcripts in sporulated oocysts and in senescent and deceased oocysts. Because many of these genes have homologues in *C. cayetanensis*, they may prove useful as biomarkers for human risk. Therefore, future studies will aim to empirically determine whether expression profiles in *E. acervulina* parallel those in human parasites.

## Supporting information

**S1 Fig. Oocysts at hour zero.** At hour 0, oocysts lack distinctive sporocysts. 100X under oil immersion.
(TIF)

**S2 Fig. Oocysts at hour 24.** At hour 24, 83% of oocysts contain distinctive sporocysts. 100X under oil immersion.
(TIF)

**S1 Table. *E. acervulina* gene oligonucleotide sequences used in this study.** Gene names and annotated protein descriptions (columns A, B), oligonucleotide sequence (column C), and suggested final concentration (nM, nanomolar) for use (column D) are included. All sequences are novel to this work.
(XLSX)

**S2 Table. RNA-Seq data for all *E. acervulina* annotated genes and biological replicates in the study.** Included are the gene name (column A), original annotated protein description (column B), protein sequence (column C), Transcripts Per Million (TPM) per biological replicate in the 24-hour time course (columns D-V), overall mean TPM (column W), and TPM standard deviation (SD, column X).
(XLSX)

**S3 Table. Summarized RNA-Seq data for the study.** Included are the *E. acervulina* annotated genes (column A), and mean TPM (columns B-H) and SD (columns I-O) for biological replicates at each time point.
(XLSX)

**S4 Table. Blast2GO results for the 53 constitutively expressed *E. acervulina* genes with >1,000 TPM.** Gene names are included in column A and the mean overall TPM throughout the 24-hour time course is in column B. BLASTP hits and annotated protein descriptions are in columns C and D, respectively. Column D includes the original *E. acervulina* genome annotation and hits to *E. acervulina*, other *Eimeria* spp., *C. cayetanensis*, *T. gondii*, and other apicomplexan species. The table does not include all BLASTP results. Other data include the overall number of hits found (column E), hit e-value (column F), the mean percent similarity for hits (column G), and length of protein amino acid sequence (H). Gene Ontology (GO) data includes number of GO terms (column I), GO ID numbers (column J), and GO term names (column K). Kyoto Encyclopedia of Genes and Genomes (KEGG) biochemical pathway

data includes enzyme codes (column L) and enzyme names (column M). Shading indicates genes for which KEGG analysis found important biochemical pathways. InterPro data include ID names (column N), GO ID numbers (column O), and GO names (column P). Bolded genes indicate those (originally annotated as hypothetical protein) where BLASTP identified more informative hits in other species. Italicized genes indicate those (originally annotated as hypothetical protein) where BLASTP found matches to hypothetical or uncharacterized protein in other species, or there was no other hit.
(XLSX)

**S5 Table. List of the 285 annotated GO terms for the constitutively expressed *E. acervulina* genes with >1,000 TPM.** GO terms (column A) are stratified by GO subontologies Biological Process (BP), Molecular Function (MF), and Cellular Component (CC). The total number of GO terms for subontology is also provided at the bottom of columns B-D.
(XLSX)

**S6 Table. *E. acervulina* generic Transcription Factor (TF)-encoding genes and Blast2GO results in *E. acervulina* and *C. cayetanensis*.** Data in the table include *E. acervulina* gene names and annotated protein descriptions (columns A, B), the number of GO terms (column C), GO ID numbers (column E), and GO term names (column F) found for the *E. acervulina* protein sequences. Also included are the top *C. cayetanensis* BLASTP hit that is annotated as a TF (but not necessarily the top overall BLAST hit) and hit e-value (column G). Columns H-N include mean TPM for biological replicates at each time point. The overall mean TPM for all samples (column O), log2 ratio differential expression between time points T24 and T0 (column P), and differential expression adjusted *p*-value (column Q) are also included.
(XLSX)

**S7 Table. *T. gondii* ApiAP2 TF-encoding genes for which we identified homologs in *E. acervulina* and *C. cayetanensis* by Blast2GO.** Data in the table include *T. gondii* gene names and protein descriptions (columns A, B), the top BLASTP hit in *E. acervulina* and hit e-value (columns C, D) and *E. acervulina* gene identifier for each hit (column E). Sometimes more than one *T. gondii* gene matched to the same *E. acervulina* gene. Also included are the number of GO terms (column F), GO ID numbers (column G), and GO term names (column H) found for the *E. acervulina* protein sequences. The ApiAP2 or other transcription factor BLASTP hit and e-value are listed for *C. cayetanensis* after Blast2GO was run on the *E. acervulina* protein sequences (columns I, J). Columns K-Q include mean TPM for biological replicates at each time point. The overall mean TPM for all samples (column R), log2 ratio differential expression between time points T24 and T0 (column S), and differential expression adjusted *p*-value (column T) are also included. Genes in bold represent unique homologs in *E. acervulina* that met our criteria for significant differential expression.
(XLSX)

**S8 Table. Differentially expressed genes between *E. acervulina* immature and mature oocysts.** DESeq2 was used in Geneious to compare expression between biological replicates of mature (T24) and immature (T0) oocysts. Data in the table include the gene identifier and protein description (columns A, B), log2 ratio between mature and immature oocysts, and adjusted *p*-value. Data are shown for all annotated *E. acervulina* genes and sorted high to low based on log2 ratio. NA indicates "not available" for genes with zero TPM among biological replicates.
(XLSX)

**S9 Table. Blast2GO results for the 8 significantly differentially expressed genes in immature (T0)** *E. acervulina* **oocysts.** Data in the table include *E. acervulina* gene names (column A) and mean TPM at T0 (column B). BLASTP hits and annotated protein descriptions are in columns C and D, respectively. Column D includes the original *E. acervulina* genome annotation and hits to *E. acervulina*, other *Eimeria* spp., *C. cayetanensis*, *T. gondii*, and other apicomplexan species. The table does not include all BLASTP results. Other data include the overall number of hits found (column E), hit e-value (column F), the mean percent similarity for BLASTP hits (column G), and length of protein amino acid sequence (H). Gene Ontology (GO) data includes number of GO terms (column I), GO ID numbers (column J), and GO term names (column K). Also included is the total number of GO terms (bottom of column I). KEGG biochemical pathway data includes enzyme codes (column L) and enzyme names (column M). Shading indicates genes for which KEGG analysis found important biochemical pathways. InterPro data include ID names (column N), GO ID numbers (column O), and GO names (column P). Bolded genes indicate *E. acervulina* genes (originally annotated as hypothetical protein) where BLASTP identified more informative hits in other species. Italicized genes indicate *E. acervulina* genes (originally annotated as hypothetical protein) where BLASTP found matches to hypothetical proteins in other species.
(XLSX)

**S10 Table. List of the 37 annotated GO terms for the constitutively expressed** *E. acervulina* **genes with >1,000 TPM.** GO terms (column A) are stratified by GO subontologies Biological Process (BP), Molecular Function (MF), and Cellular Component (CC). The total number of GO terms for subontology is also provided at the bottom of columns B-D.
(XLSX)

**S11 Table. Blast2GO results for the 56 significantly differentially expressed genes in mature (T24)** *E. acervulina* **oocysts.** Data in the table include *E. acervulina* gene names (column A) and mean TPM at T24 (column B). Also included are the mean TPM at T24. BLASTP hits and annotated protein descriptions are in columns C and D, respectively. Column D includes the original *E. acervulina* genome annotation and hits to *E. acervulina*, other *Eimeria* spp., *C. cayetanensis*, *T. gondii*, and other apicomplexan species. The table does not include all BLASTP results. Other data include the overall number of hits found (column E), hit e-value (column F), the mean percent similarity for hits (column G), and length of protein amino acid sequence (H). Gene Ontology (GO) data includes number of GO terms (column I), GO ID numbers (column J), and GO term names (column K). KEGG biochemical pathway data includes enzyme codes (column L) and enzyme names (column M). Shading indicates genes for which KEGG analysis found important biochemical pathways. InterPro data include ID names (column N), GO ID numbers (column O), and GO names (column P). Bolded genes indicate *E. acervulina* genes (originally annotated as hypothetical protein) where BLASTP identified more informative hits in other species. Italicized genes indicate *E. acervulina* genes (originally annotated as hypothetical protein) where BLASTP found matches to hypothetical proteins in other species or there were no other hits.
(XLSX)

**S12 Table. List of the 215 annotated GO terms for the constitutively expressed** *E. acervulina* **genes with >1,000 TPM.** GO terms (column A) are stratified by GO subontologies Biological Process (BP), Molecular Function (MF), and Cellular Component (CC). The total number of GO terms for subontology is also provided at the bottom of columns B-D.
(XLSX)

## Acknowledgments

We thank SeonWoo Kim for technical support with NextSeq operation. We are also grateful to Peter Thompson for RNA-Seq advisement and sequencing assistance.

## Author Contributions

**Conceptualization:** Matthew S. Tucker, Mark C. Jenkins, Benjamin M. Rosenthal.

**Data curation:** Matthew S. Tucker, Mark C. Jenkins, Benjamin M. Rosenthal.

**Formal analysis:** Matthew S. Tucker.

**Investigation:** Matthew S. Tucker, Celia N. O'Brien, Mark C. Jenkins, Benjamin M. Rosenthal.

**Methodology:** Matthew S. Tucker, Celia N. O'Brien, Mark C. Jenkins.

**Project administration:** Benjamin M. Rosenthal.

**Supervision:** Benjamin M. Rosenthal.

**Visualization:** Matthew S. Tucker.

**Writing – original draft:** Matthew S. Tucker, Benjamin M. Rosenthal.

**Writing – review & editing:** Matthew S. Tucker, Celia N. O'Brien, Mark C. Jenkins, Benjamin M. Rosenthal.

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
