## [Decision Letter · Decision Letter 0]

31 Aug 2021

PONE-D-21-23152

Dynamically expressed genes provide candidate viability biomarkers in a model coccidian

PLOS ONE

Dear Dr. Benjamin M. Rosenthal,

Thank you for submitting your manuscript to PLOS ONE. After careful consideration, we feel that it has merit but does not fully meet PLOS ONE’s publication criteria as it currently stands. Therefore, we invite you to submit a revised version of the manuscript that addresses the points raised during the review process.

This manuscript is an interesting study but there are main two issues about the study need clarification from the authors to the reviewers comments; why did the study  following of oocysts sporulation only 24 hours? And the second one, What happens to the transcripts old, deteriorate, and eventually died oocysts?

We look forward to receiving your revised manuscript.

Kind regards,

Shawky M. Aboelhadid, PhD

Academic Editor

PLOS ONE

Journal Requirements:

2. In your Methods section, please include a comment about the state of the animals following this research. Were they euthanized or housed for use in further research? If any animals were sacrificed by the authors, please include the method of euthanasia and describe any efforts that were undertaken to reduce animal suffering.

 [This work was supported by USDA Projects “Detection and Control of Foodborne Parasites for Food Safety” 8042-32000-113-00D and “Development of Control and Intervention Strategies for Avian Coccidiosis” 8042-32000-111-00-D.]

5. Please amend either the title on the online submission form (via Edit Submission) or the title in the manuscript so that they are identical.

Reviewers' comments:

Reviewer's Responses to Questions

**Comments to the Author**

1. Is the manuscript technically sound, and do the data support the conclusions?

Reviewer #1: Yes

Reviewer #2: Yes

2. Has the statistical analysis been performed appropriately and rigorously? 

Reviewer #1: Yes

Reviewer #2: Yes

3. Have the authors made all data underlying the findings in their manuscript fully available?

Reviewer #1: Yes

Reviewer #2: Yes

4. Is the manuscript presented in an intelligible fashion and written in standard English?

Reviewer #1: Yes

Reviewer #2: Yes

5. Review Comments to the Author

Reviewer #1: In this manuscript authors report their findings from a genome wide RNA-Seq study done using Eimeria acervulina oocysts to determine the biological processes involve in oocyst maturation in this organism. They also propose E. acervuline as a surrogate for Cyclospora cayetanensis, a related apicomplexan causing human disease. The data reported in the manuscript may provide a framework for a) understanding the biology of oocysts sporulation in Eimeria and related genus, such as Cyclospora, b) discovery of new biomarkers that can be used for determination of viability, and developmental stages, such as sporulated and unsporulated stages, in these organisms.

The manuscript is well written, and well organized.

Major comments:

- Authors do not explain why the first 24 h of sporulation process is analyzed in this study. The question here is that; does the first 24 hour cover the whole sporulation process for E. acervulina? Please explain in the manuscript.

- Authors state that “ At each time point, oocysts were centrifuged at 3,500 rpm for 5 min at 4°C and washed with deionized water to remove excess potassium dichromate. T0 and T24 oocysts were examined at 400X with an Axio microscope (Zeiss, Germany). Images were captured using a Zeiss AxioScope camera and AxioVision imaging software.” Line 145-149. What was the sporulation rate at the 24h time point (and at other time points, if data is available)? Also representative images at different time points should be made available as supplementary data.

- Sporulated/mature, and un-sporulated/immature oocysts terms were used interchangeably in different parts of the manuscript. I recommend to use either sporulated- un-sporulated, or immature- mature terms throughout the manuscript, unless authors assign different meaning to these terms. If so, please explain.

Minor corrections & comments:

- Lines 317-318, change to “We used Transcripts Per Million (TPM), as normalization method, to summarize expression of the 6,867 annotated genes.

- Line 323, change to “-during this time interval”.

- Line 339, do you mean “Only ~600-900 genes (<15% of the genes) in each time point were expressed >100 TPM (Table 2)?

- Line 339-340, “To better understand global expression, we conducted an analysis of transcriptional bias at each time point.” Please clearly explain what you mean by “transcriptional bias”.

- Line 787, Please change to “--Cyclospora may constitute—“. Cyclospora and Eimeria are separate genus under Eimeriidae family, under current scientific classification.

- Lines 1016-1020, “We identified four genes (EAH_00007450, EAH_00054340, EAH_00057900, and EAH_00067850) that steadily increased in expression in maturing oocysts and encode enzymes involved in major energy and metabolic pathways. These enzymes (glyceraldehyde-3-phosphate dehydrogenase, LDH, Fructose-bisphosphate aldolases, respectively)”. 4 genes, 3 enzymes, respectively. Which 3 genes encodes 3 enzymes, clarify.

- Line 1101, “…and GRAs are…” Please enter open name[dense granule antigen?/protein].

- Line 1314, “…and strains of C. cayetanensis…” Please use the term “isolates” instead of “ strains”, because C. cayetanensis is uncultarable, strains cannot be defined.

Reviewer #2: The manuscript submitted by Tucker et al describes a detailed RNAseq time course study of sporulation in Eimeria acervulina oocysts. The work is detailed and well referenced, adding to the body of knowledge for Eimeria. Extrapolation to Cyclospora is topical and relevant.

Major comments

1. The work very clearly presents detailed transcript variation between immature (unsporulated, non-infectious) and mature (sporulated, infectious) oocysts, highlighting several candidate biomarkers for sporulation. However, detection of transcripts associated with completion of sporulation is not a clear biomarker for viability or ongoing infectious risk. What happens to these transcripts as oocysts age, deteriorate, and eventually die? The authors do eventually touch on this, but comments in prominent sections of the manuscript such as the abstract should reflect this key point.

Minor comments

1. Line 33 and below. While some transcripts are found to be expressed throughout the 24 h sampling window of oocyst maturation, I don’t think these should be called ‘constitutive’, since the rest of the lifecycle is not considered. For example, an oocyst wall protein would not be constitutively transcribed.

2. Line 63-64. The toxin producing C. perfingens strains that cause NE in chickens are not usually the same that cause disease in humans. Suggest remove text after reference [2].

3. Line 81. 10% of what? - People, presumably, rather than the foodstuffs.

4. Lines 96-97. Sporozoite release requires stimulation - it is not inevitable after oocyst rupture.

5. Line 136 and below. Centrifuge speeds should be given in g, not rpm, or give rotor diameter.

6. Line 140. What concentration(s) were the oocysts sporulated in?

7. Table 3. Can the authors add a column for T12? This can improve characterization per transcript.

8. The discussion is detailed and well referenced, but it is very long with some repetition from the results - 12 pages is a very long discussion. Can this be abbreviated?

6. PLOS authors have the option to publish the peer review history of their article (what does this mean?). If published, this will include your full peer review and any attached files.

Reviewer #1: No

Reviewer #2: No

---

## [Author Response · Author response to Decision Letter 0]

15 Sep 2021

We thank the reviewers for their careful consideration of the manuscript. Please see the attached "response to reviewers" for a detailed description of how we have improved the manuscript according to each of their observations and questions.

---

## [Decision Letter · Decision Letter 1]

20 Sep 2021

Dynamically expressed genes provide candidate viability biomarkers in a model coccidian

PONE-D-21-23152R1

Dear Dr. Benjamin M. Rosenthal,

We’re pleased to inform you that your manuscript has been judged scientifically suitable for publication and will be formally accepted for publication once it meets all outstanding technical requirements.

Kind regards,

Shawky M. Aboelhadid, PhD

Academic Editor

PLOS ONE

Additional Editor Comments (optional):

Reviewers' comments:

Reviewer's Responses to Questions

**Comments to the Author**

1. If the authors have adequately addressed your comments raised in a previous round of review and you feel that this manuscript is now acceptable for publication, you may indicate that here to bypass the “Comments to the Author” section, enter your conflict of interest statement in the “Confidential to Editor” section, and submit your "Accept" recommendation.

Reviewer #1: All comments have been addressed

Reviewer #2: All comments have been addressed

2. Is the manuscript technically sound, and do the data support the conclusions?

Reviewer #1: Yes

Reviewer #2: Yes

3. Has the statistical analysis been performed appropriately and rigorously? 

Reviewer #1: Yes

Reviewer #2: Yes

4. Have the authors made all data underlying the findings in their manuscript fully available?

Reviewer #1: Yes

Reviewer #2: Yes

5. Is the manuscript presented in an intelligible fashion and written in standard English?

Reviewer #1: Yes

Reviewer #2: Yes

6. Review Comments to the Author

Reviewer #1: (No Response)

Reviewer #2: Thanks for the revisions - no further comments.

For some reason there is a minimum character limit here, so writing this as well...!

7. PLOS authors have the option to publish the peer review history of their article (what does this mean?). If published, this will include your full peer review and any attached files.

Reviewer #1: No

Reviewer #2: No

---

## [Editor Report · Acceptance letter]

24 Sep 2021

PONE-D-21-23152R1 

Dynamically expressed genes provide candidate viability biomarkers in a model coccidian 

Dear Dr. Rosenthal:

I'm pleased to inform you that your manuscript has been deemed suitable for publication in PLOS ONE. Congratulations! Your manuscript is now with our production department. 

Kind regards, 

on behalf of

Professor Shawky M. Aboelhadid 

Academic Editor

PLOS ONE